# Modelling onchocerciasis-associated epilepsy and the impact of ivermectin treatment on its prevalence and incidence

Jacob N. Stapley [1,2] ✉, Jonathan I. D. Hamley [1,2,3,4], Martin Walker [1,2,5], Matthew A. Dixon [1,2], Robert Colebunders [6] & Maria-Gloria Basáñez [1,2] ✉

Retrospective cohort studies in Cameroon found an association between *Onchocerca volvulus* microfilarial load in childhood (measured in 1991–1993) and risk of developing epilepsy later in life (measured in 2017). We parameterised and integrated this relationship (across children aged 3–15 years) into the previously published, stochastic transmission model, EPIONCHO-IBM, for *Simulium damnosum* sensu lato-transmitted onchocerciasis. We simulated 19 years (1998–2017) of annual ivermectin mass drug administration (MDA) reflecting coverage in the study area, and modelled epilepsy prevalence and incidence. Scenario-based simulations of 25 years of (annual and biannual) MDA in hyper- and holoendemic settings, with 65% and 80% therapeutic coverage, were also conducted. EPIONCHO-IBM predicted 7.6% epilepsy prevalence (compared to 8.2% in the Cameroon study) and incidence of 317 cases/100,000 person-years (compared to 350). In hyperendemic areas, 25 years of biannual MDA (80% coverage) eliminated onchocerciasis-associated epilepsy (OAE) and protected untreated under-fives from its development. Strengthening onchocerciasis programmes, implementing alternative strategies, and evaluating treatment for under-fives and school-age children are crucial to prevent OAE in highly-endemic settings.

Human onchocerciasis, caused by the filarial nematode *Onchocerca volvulus*, is endemic in 27 countries of sub-Saharan Africa (SSA)[1]. The disease is transmitted among humans through the bites of blackfly (*Simulium*) vectors[2]. The Global Burden of Disease (GBD) study estimated 1.23 million disability-adjusted life-years due to onchocerciasis in 2019[3]. Major control programmes in SSA comprised the Onchocerciasis Control Programme in West Africa (OCP, 1975–2002) and the African Programme for Onchocerciasis Control (APOC, 1995–2015)[4]. Currently, the main intervention strategy is annual mass drug administration (MDA) of ivermectin.

Untreated onchocerciasis is associated with substantial morbidity, including visual impairment leading to blindness and skin disease, and is also responsible for excess mortality (above that caused by blindness), particularly in the young[5,6]. Epidemiological studies have indicated an association between onchocerciasis and epilepsy[6–8]. In particular, the systematic review and meta-analysis presented by Pion et al.[7] evaluated the relationship between onchocerciasis prevalence and that of epilepsy at the community level for studies distributed from West to East Africa[7]. Onchocerciasis-associated epilepsy (OAE) mainly occurs in highly endemic regions with intense ongoing transmission.

[1]MRC Centre for Global Infectious Disease Analysis, Department of Infectious Disease Epidemiology, School of Public Health, Imperial College London, London, UK. [2]London Centre for Neglected Tropical Disease Research, Department of Infectious Disease Epidemiology, School of Public Health, Imperial College London, London, UK. [3]Department of Visceral Surgery and Medicine, Inselspital, Bern University Hospital, University of Bern, Bern, Switzerland. [4]Multidisciplinary Center for Infectious Diseases, University of Bern, Bern, Switzerland. [5]Department of Pathobiology and Population Sciences, Royal Veterinary College, Hatfield, UK. [6]Global Health Institute, University of Antwerp, Antwerp, Belgium. ✉e-mail: j.stapley20@imperial.ac.uk; m.basanez@imperial.ac.uk

**Table 1 | Comparison Between Observed and Modelled Baseline Epidemiological and Transmission Conditions**

| Study | Mean Mf Prevalence (Range) (%) | Mean CMFL (Range) (mf/ss) | Mean ATP (Range) (L3/person/year) | Mean ABR (Range) (bites/person/year) | Mean L3 Load (Range) (L3/fly) |
|---|---|---|---|---|---|
| Pion et al.[14] | 86.7 (80.9 – 96.8) | 50.6 (13.9 – 136.2) | – | – | – |
| Barbazan et al.[15] | – | – | 1583[a] (184 – 3113) | 39,765 (13,647 – 98,028) | 0.0398 (0.0042 – 0.0426) |
| EPIONCHO-IBM[b] | 88.9 (83.3 – 91.3) | 104.6 (79.7 – 121.4) | 1700 (550 – 4170) | 41,922 (14,000 – 100,000) | 0.0406 (0.0393 – 0.0417) |

*Mf* Prevalence microfilarial prevalence; *CMFL* Community Microfilarial Load; *ABR* annual biting rate; *ATP* annual transmission potential; *L3/fly* no. of infective larvae per fly.

[a]Mean ATP at Ngoro[15].

[b]Model outputs for EPIONCHO-IBM with ABR = 41,992 and with ABR = 14,000 and 100,000 (in brackets) to reflect minimum and maximum values as reported by ref. 15.

The data presented in Pion et al.[14] were collected in 1991–1993 in the areas of Bitang, Yambassa and Yébékolo (see Supplementary Fig. S5, which reproduces the study area of ref. 10); the data presented in Barbazan et al.[15] were collected in 1993–1994 in Ngoro and Bokito (also in Supplementary Fig. S5).

A defining characteristic of OAE is its onset between 3–18 years of age in previously healthy children, presenting a spectrum of conditions ranging from seizures to nodding and Nakalanga syndromes[6]. OAE has mainly been reported in areas of high mesoendemicity and hyperendemicity for onchocerciasis, and recedes upon intensification of control[8,9].

The retrospective cohort studies of Chesnais et al.[10,11] provide the strongest evidence of temporality in the relationship between past *O. volvulus* microfilarial (mf) load and epilepsy incidence. Villagers who, aged between 5–10 years[10] or 5–15 years[11] had been assessed for mf load were re-visited 25–30 years later to investigate the incidence of epilepsy in the intervening period, indicating a dose-response relationship between mf load in childhood and the risk of developing epilepsy later in life.

Previously, we investigated the effect of ivermectin MDA on blindness incidence and excess mortality using a deterministic version of the EPIONCHO transmission model[12]. There is renewed interest in understanding the consequences of large-scale onchocerciasis control and elimination programmes on disease burden in SSA, expanding the range of sequelae beyond ocular and skin disease thus far included in the GBD Study[3]. Therefore, we incorporated OAE into our stochastic EPIONCHO-IBM transmission model[13] to: (i) test in silico the dose-response relationship between *O. volvulus* mf load in childhood and the probability of developing epilepsy later in life as crucial to the process by which onchocerciasis is associated with epilepsy; (ii) validate the model by testing its ability to capture the results presented in ref. 10 as well as more broadly in ref. 7, and (iii) investigate the effect of long-term ivermectin MDA on predicted prevalence and incidence of epilepsy under a range of epidemiological and programmatic scenarios.

## Results

### Calibration of EPIONCHO-IBM

An estimated vector annual biting rate (ABR) = 41,922 bites/person/year was found to be the best-fit biting rate estimate capable of generating mf loads consistent with those recorded in 1991–93 in ref. 10. Supplementary Table S1 presents measures of central tendency and variation in mf loads for children aged 5–10 years for the 729 individuals who were examined in 1991–93 and assessed for epilepsy in 2017 compared with the same metrics derived from EPIONCHO-IBM.

Table 1 compares the epidemiological and entomological results recorded in the Mbam Valley at baseline[14,15], with EPIONCHO-IBM-generated values for the model calibrated with ABR = 41,922, indicating that model outputs are consistent with observations. The ranges around the modelled values correspond to ABR of 14,000 and 100,000 by way of representing uncertainty in the transmission conditions in the study area[15].

Supplementary Figure S1 presents the observed age-specific mf loads for children aged 5–10 years in 1991–93, their 95% confidence intervals (95% CI), the modelled mf loads for ABR = 41,922 and the corresponding results for ABR = 14,000 and ABR = 100,000.

### Dose-response relationship between microfilarial load and epilepsy

The dose-response relationship between mf load and the onset probability of OAE incorporated into the model is shown in Fig. 1.

### Effect on OAE prevalence and incidence of 19 years of annual community-directed treatment with ivermectin (CDTI)

The prevalence (and 95% CI) of epilepsy reported in ref. 10 for those aged 30–35 years (after 19 rounds of annual CDTI) was 8.2% (95% CI = 6.5–10.5%); 9.5% (7.0–13.0%) in males and 6.8% (4.6–9.9%) in females, compared to 7.6% generated by the model (8.6% in males and 6.7% in females). The reported incidence was 350 cases/100,000 person-years[10], compared to 317 generated by EPIONCHO-IBM

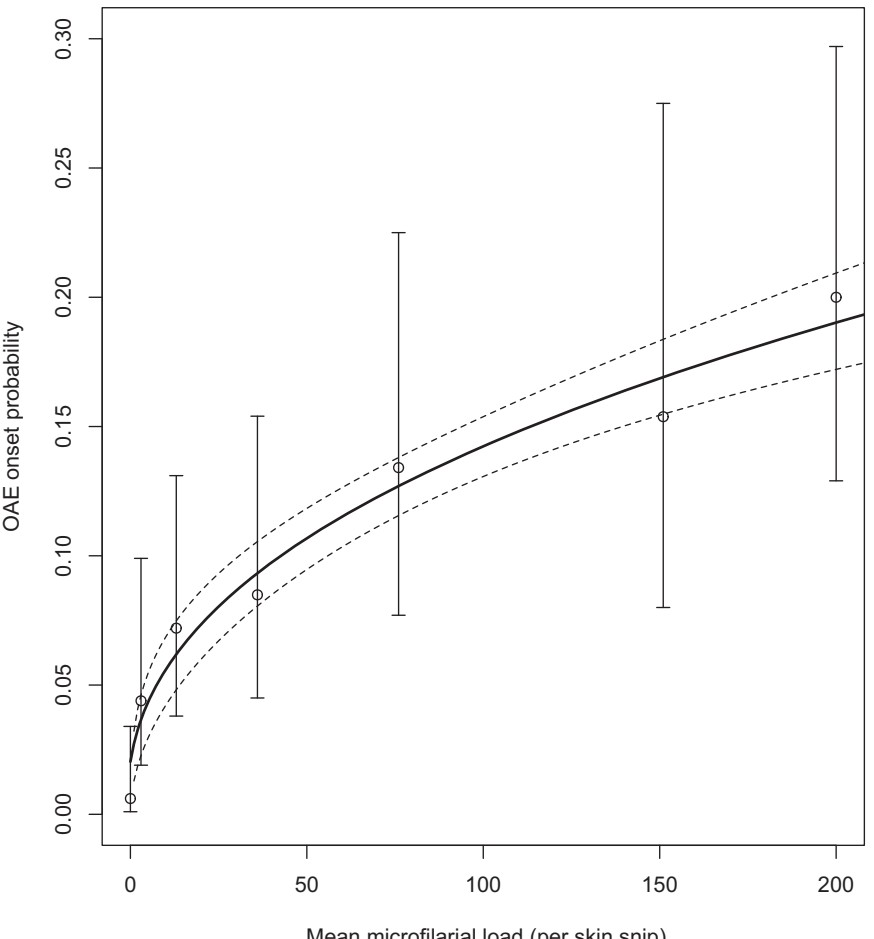

**Fig. 1 | Probability of OAE onset as a function of microfilarial (mf) load.** Data points ($n$ = 729) (circles) are OAE onset probabilities (taken as the proportion of individuals with epilepsy) plotted for the midpoints of the seven binned (mean) mf loads from Chesnais et al.[10], and error bars associated with the data points are the 95% confidence intervals reported in ref. 10 (reproduced in Supplementary Table S5). A non-linear function (Eq. 1) in the form of a generalised linear model (GLM) with a log link was fitted to the data (solid black line), with dashed lines around the solid line representing the 95% confidence limits associated with the model fit. The values of the parameters in Eq. 1 were estimated as $\beta_0$ = −3.89 and $\beta_1$ = 0.42.

(Supplementary Table S2). Figure 2 shows the prevalence of epilepsy at baseline (1991–1992) in the village of Nyamongo (6.4%) and after 19 years of annual CDTI (5.1%), a 20.3% reduction[16], compared to EPIONCHO-IBM-predicted values of 6.1% and 4.9%, respectively, a 19.7% reduction.

### Sensitivity analyses to pre-intervention conditions

Figure 3 shows the relationship between modelled OAE prevalence (all ages, by sex) and ABR. OAE prevalence is predicted to be higher in males than females and to increase non-linearly with ABR.

Figure 4 shows the relationship between OAE prevalence and mf prevalence generated by EPIONCHO-IBM for three values of inter-individual exposure heterogeneity (see "Methods" and Supplementary Table S3) and compares this relationship with that fitted by Pion et al.[7] to their collated data across SSA using logistic meta-regression. The results of EPIONCHO-IBM fall within the 95% CI of the statistical model[7] (Fig. 4a) and capture the patterns in the data (Fig. 4b).

### Scenario modelling of the impact of ivermectin treatment on OAE

The effect of 25 years of annual ivermectin MDA at minimal (65% of total population, 5% systematic non-adherence, SNA) and enhanced coverage (80% and 1%, respectively), for two (hyper- and holo-) ende-micity levels, on OAE prevalence and incidence (no. of cases per

100,000 person-years averaged over every 5 years) is presented in Fig. 5a, b respectively.

For the hyperendemic setting (60% mf prevalence at baseline), the modelled reduction in OAE prevalence during the 25-year treatment period is approximately 40% for both the minimal and enhanced coverage levels, but for the latter, a greater reduction in OAE pre-valence is predicted after treatment cessation, with a maximal reduc-tion of over 70% 45 years post-CDTI, such that OAE prevalence does not reach pre-intervention levels during the 100-year simulated per-iod. For the minimal coverage scenario, the reduction in OAE pre-valence also continues after treatment cessation, but its magnitude is less, with a maximal reduction of 45% 15 years post-CDTI and a trend to bounce back to pre-CDTI levels (Fig. 5a).

For the holoendemic setting (80% mf prevalence at baseline), annual CDTI effects a reduction in OAE prevalence of ~30% in both coverage scenarios. Like in the hyperendemic setting, OAE prevalence continues decreasing after treatment cessation, albeit for only 5 years, with both minimal and enhanced coverage simulations bouncing back to pre-intervention levels (Fig. 5a). In the hyperendemic setting, the maximal reduction is 78% under minimal coverage and 99% under enhanced coverage, whereas for the holoendemic setting these values are 58% and 68%, respectively. The incidence does not return to initial levels during the following 75 years post-CDTI for the enhanced cov-erage scenario in the hyperendemic setting. For the minimal coverage

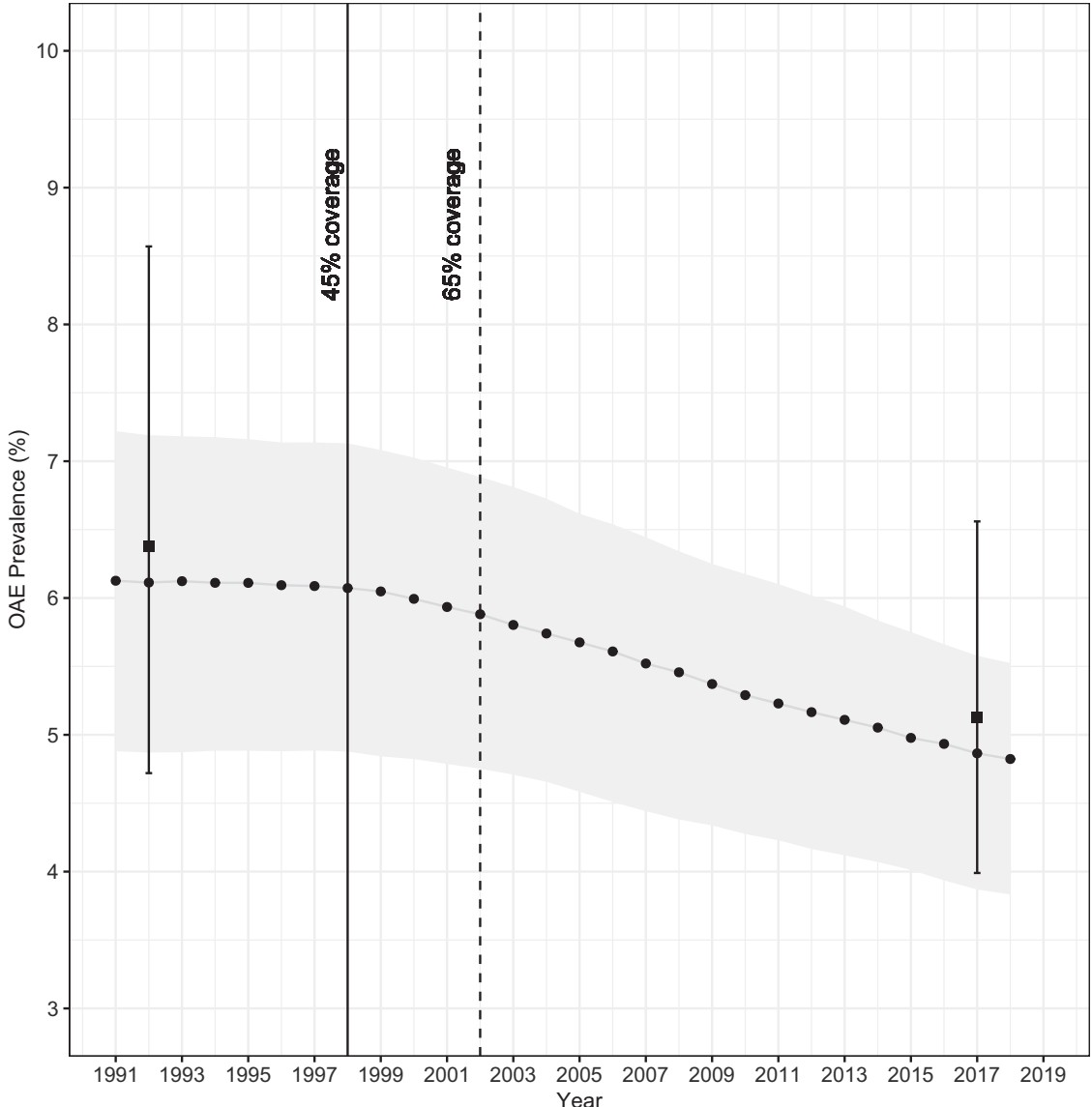

**Fig. 2 | Observed and EPIONCHO-IBM-predicted prevalence of epilepsy in Nyamongo, Mbam Valley, Cameroon at baseline and after 19 years of annual community-directed treatment with ivermectin (CDTI).** The squares are the reported prevalence (n = 627 in 1991–1992, and n = 1151 in 2017) from Boullé et al.[16], and the error bars are the 95% confidence intervals (95% CIs) around the prevalence. The epilepsy prevalence in 1991–92 was 6.4% (95% CI = 4.7–8.6%), and in 2017 it was 5.1% (95% CI = 4.0–6.6%). The grey line with black circles represents the OAE prevalence trend predicted by EPIONCHO-IBM for annual biting rate (ABR) = 41,922 bites/person/year; the grey shaded area indicates modelled uncertainty around OAE prevalence when simulating a minimum ABR = 14,000 and a maximum of 100,000[15]. All simulations were conducted with 300 model repeats. The solid black vertical line indicates the start of annual CDTI at 45% coverage; the vertical dashed line indicates the increase in coverage to 65% (of the total population). Therapeutic coverage levels are informed by Kamga et al.[19].

scenario, and for both coverage scenarios in the holoendemic setting, incidence bounces back after treatment cessation (Fig. 5b).

Supplementary Figure S2 presents the trends in OAE prevalence and incidence under biannual (6-monthly) CDTI. Notably, in the hyperendemic setting, OAE is nearly eliminated under enhanced coverage. In the holoendemic setting, the maximal incidence reduction is 72% under minimal coverage and 84% under enhanced coverage; twenty-five years of 6-monthly treatment are not sufficient to prevent OAE from returning to pre-intervention levels.

The contribution to OAE incidence of (untreated) children under 5 is presented in Supplementary Fig. S3 for the hyper- and holoendemic settings and treatment frequencies investigated. In hyperendemic settings, untreated children contribute an incidence of approximately 6 cases/100,000 persons/year (roughly 10% of the total incidence); the value for holoendemic settings is approximately 13 cases/100,000 persons/year (12% of the total incidence). (In the model, only those aged 3–15 years contribute to OAE incidence.) Because transmission intensity decreases during the treatment period, the contribution of the under-fives decreases, to increase again once treatment stops, with the exception of biannual CDTI with enhanced coverage for the hyperendemic setting. This regimen essentially protects untreated children from developing OAE.

## Discussion

We incorporated OAE into EPIONCHO-IBM using the dose-response relationship between *O. volvulus* mf load in childhood and the probability of developing epilepsy later in life reported in ref. 10 (Fig. 1) and used a 3–15-year age range for the onset of OAE[10,11,17]. The EPIONCHO-IBM-modelled relationship between ABR and epilepsy prevalence was strongly non-linear (Fig. 3) and consistent with the range of epilepsy

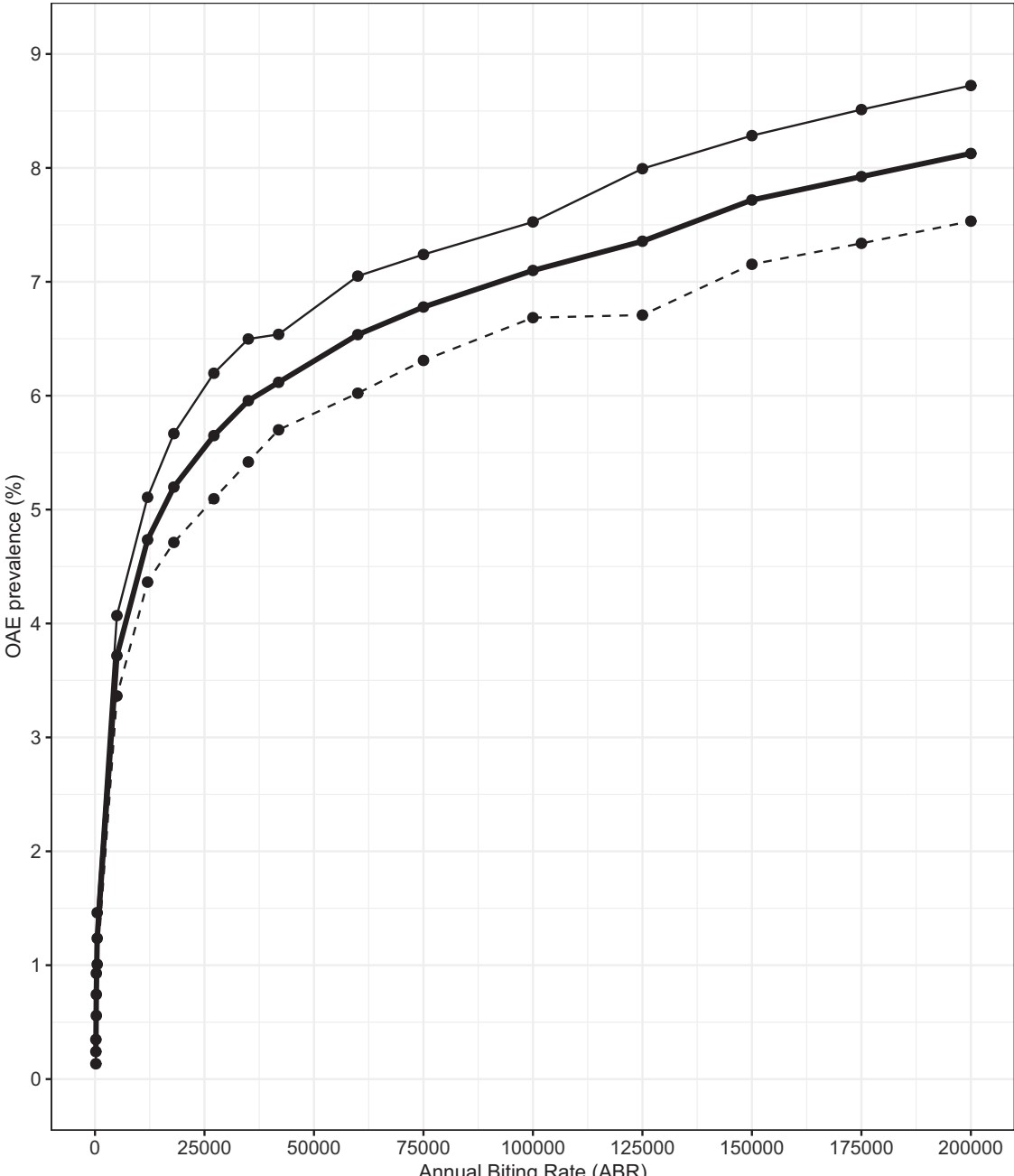

**Fig. 3 | EPIONCHO-IBM modelled onchocerciasis-associated epilepsy (OAE) prevalence vs. Annual Biting Rate (ABR).** The solid thick black line represents OAE prevalence for all ages in the total population. The solid, thin, and dashed lines indicate males and females, respectively. The ABR was varied between 200 and 200,000 bites/person/year; the value of the inter-individual exposure heterogeneity was set at $k_E = 0.3$[13]. All simulations were conducted with 300 model repeats.

prevalence reported for SSA[18]. The higher epilepsy prevalence for males compared to females in Fig. 3 results from the age- and sex-exposure functions in EPIONCHO-IBM (Supplementary Text S1 and Supplementary Fig. S4). For the 3–15-year age group (contributing to OAE onset in our model), boys are more highly exposed than girls, with the exposure of both being greater than 0 at birth. This enables the model to capture the high mf loads reported in ref. 10 for children aged 5–10 years (Supplementary Tables S1, S4) and to generate a modelled epilepsy prevalence in the 30–35-year age group (the one participating in the epilepsy survey conducted in ref. 10, 25 years after collection of parasitological data in 1991–93) of 8.6% in males and 6.7% in females (compared to the observed 9.5% in males and 6.8% in females) after accounting, in the model, for the 19 years of CDTI

experienced in the study area. According to ref. 10, the difference in epilepsy prevalence between the sexes was not statistically significant, and in our model, we do not assume a differential susceptibility to developing OAE according to sex; therefore, our results are due to EPIONCHO-IBM exposure functions. These functions had been derived from fitting an age- and sex-structured precursor of our model to data on mf infection profiles from Cameroon, as described in Supplementary Text S1.

The modelled epilepsy prevalence as a function of *O. volvulus* mf prevalence showed remarkable agreement between the results of our full-(stochastic) transmission model and those of the statistical model fitted in ref. 7 (Fig. 4). By calibrating the model with age-specific mf data for children aged 5–10 years at baseline (1991–93)[10], and simulating the

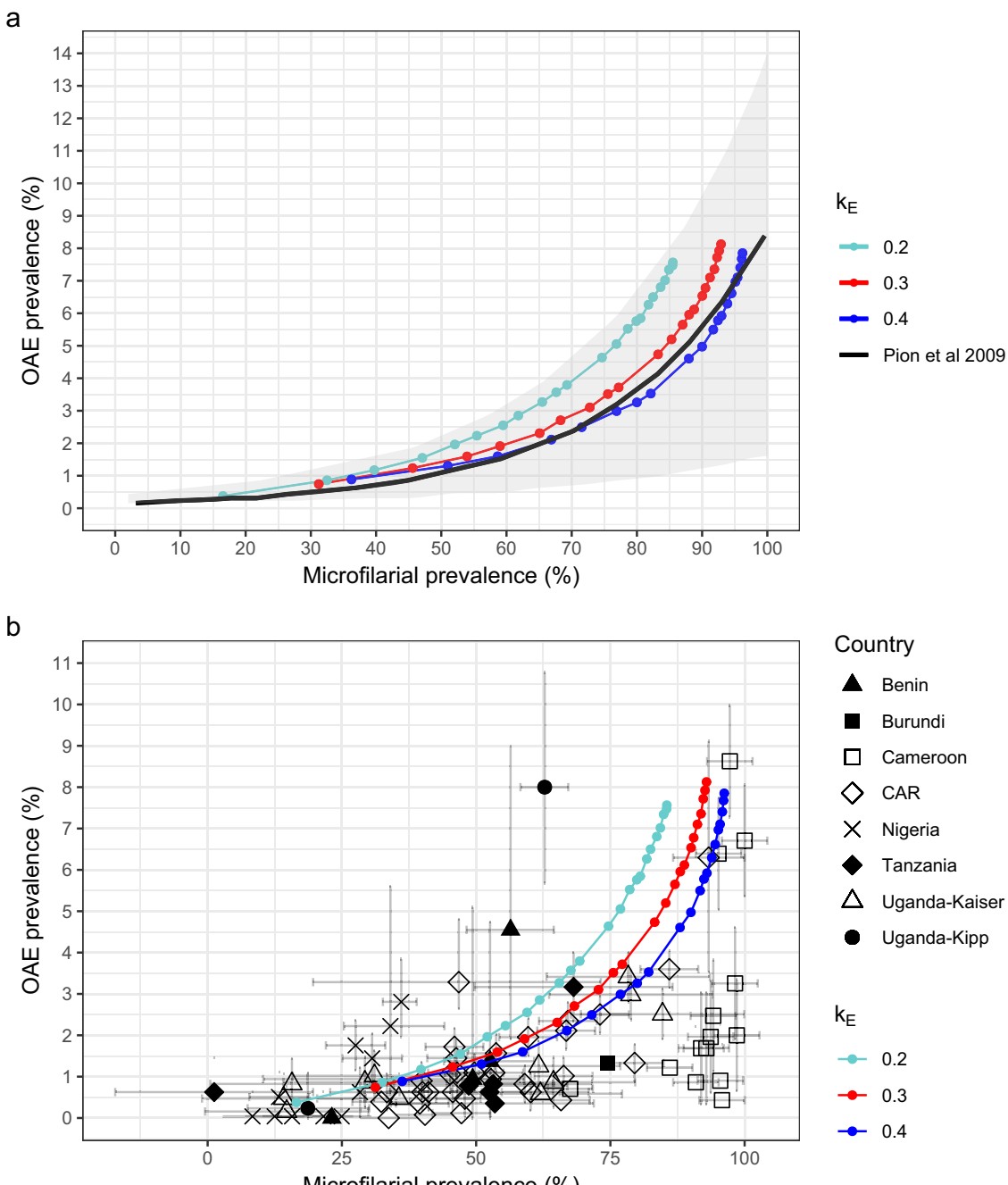

**Fig. 4 | Onchocerciasis-associated epilepsy (OAE) prevalence as a function of** *Onchocerca volvulus* **microfilarial (mf) prevalence for three values of inter-individual exposure heterogeneity.** The coloured (beaded) lines correspond to the EPIONCHO-IBM-predicted OAE prevalence. **a** The solid black line and shaded grey area represent, respectively, the logistic meta-regression model fitted by Pion et al.[7] to paired epilepsy and mf prevalence data from African settings and associated 95% confidence interval[7]. **b** The markers and error bars showing the 95% confidence intervals, represent data (sample sizes given in ref. [7]) for 7 countries across sub-Saharan Africa to which the meta-regression model in ref. [7] was fitted (CAR, Central African Republic; Uganda, two studies: Kaiser et al. (1996) and Kipp et al. (1994)[7]). The starting point of mf prevalence for each coloured line corresponds to the modelled threshold biting rate (minimum annual biting rate for onchocerciasis endemicity) for each value of the $k_E$ parameter (- 75; - 180, and - 350 bites/person/year, respectively, $k_E$ = 0.2, 0.3 and 0.4). All simulations were conducted with 300 model repeats.

CDTI history of the study area[19], EPIONCHO-IBM was able to reproduce pre-control parasitological and entomological conditions[14,15] (Table 1), as well as the epilepsy prevalence and incidence values reported for the Mbam Valley in the retrospective cohort study[10] (Supplementary Table S2) and cross-sectional study[16] (Fig. 2), after 19 years of annual CDTI. An entomological study in the Mbam Valley (2016–2017), approximately at the same time as the studies in refs. 10,16, reported annual transmission potentials (ATPs) of 0–4488 L3/person/year (mean

of 1738[20]), very similar to our modelled baseline ATP (1700, Table 1), suggesting intense ongoing transmission in this highly holoendemic area despite prolonged CDTI. Also, in 2017, IgG4 seropositivity to the *O. volvulus* Ov16 antigen (an indicator of exposure) in 7–10-year-olds was 42–55%[21]. These results highlight the importance of improving programmatic performance and deploying alternative and/or complementary interventions in highly endemic areas if onchocerciasis transmission and OAE are to be eliminated or considerably reduced.

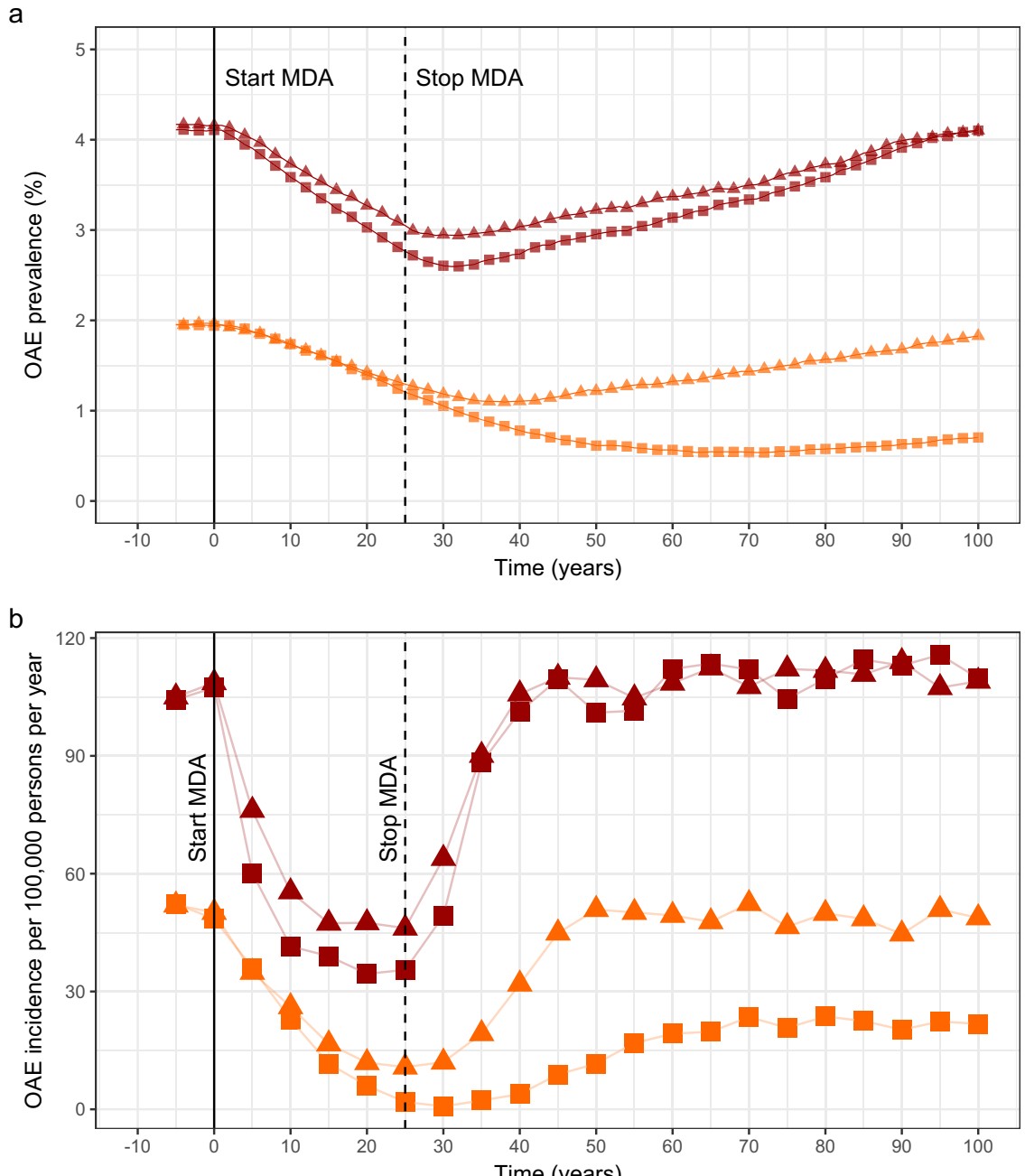

**Fig. 5 | The modelled impact of 25 years of annual community-directed treatment with ivermectin (CDTI) on onchocerciasis-associated epilepsy (OAE) for two levels of endemicity and therapeutic coverage. a** Prevalence of OAE in the overall population (all ages). **b** Incidence of OAE (no. of cases/100,000 persons/year). Orange and dark red colours denote hyper- and holoendemic settings, respectively (annual biting rate (ABR) = 1000 for 60% microfilarial (mf) prevalence; ABR = 7300 for 80% mf prevalence). Triangles and squares represent minimal and enhanced coverage, respectively. Minimal coverage: 65% therapeutic coverage of total population and 5% systematic non-adherence (SNA); enhanced coverage: 80% therapeutic coverage of total population and 1% SNA. All simulations were conducted with 300 model repeats.

The scenario-based simulation results indicated that 25 years of CDTI would lead to substantial reductions in OAE prevalence and incidence in hyperendemic settings under biannual treatment with 80% coverage of the total population. Under these programmatic conditions, the number of new cases declined to near 0 and did not increase in the following 75 years after treatment cessation (Supplementary Fig. S2). In holoendemic settings, annual or biannual treatment could reduce incidence by 60–80%, but premature treatment cessation would lead to the resurgence of transmission and OAE, as the modelled strategies would not lead to interruption of transmission (consistent with the observations of refs. 20,21 for annual CDTI). Of

note, the ABR of 7300, used to model 80% mf prevalence (Fig. 5), is notably lower than the value of c.42,000 necessary to capture the (highly holoendemic) baseline situation in the Mbam Valley (nearly 90% mf prevalence), owing to the strongly non-linear relationship between ABR and mf prevalence at endemic equilibrium in EPIONCHO-IBM[13].

Our model predicts that untreated children (under-fives, specifically those aged 3–4 years) contribute 10–12% of the total OAE incidence prior to CDTI, becoming partially or totally protected by transmission reductions resulting from the treatment of eligibles (particularly from enhanced-coverage biannual treatment in the

hyperendemic setting). Our results support the notion that OAE is preventable by strengthening onchocerciasis control and elimination efforts[17].

## Limitations

As the only driver of OAE in our model is the mf load in childhood (3–15 years), we have not accounted for differences in susceptibility due to other factors. However, the risk of developing epilepsy later in life may be more likely for younger children[10,11]. Furthermore, genetic, immunological and co-infection factors may render some individuals more inherently susceptible to developing epilepsy[10]. Besides, our model was parameterised with *Simulium damnosum* sensu lato as the vector, but in Uganda, where OAE was first reported in Africa[22], transmission was due to *S. neavei*[23]. Therefore, it cannot be ruled out that the transmission characteristics of different vector groups may influence OAE patterns (but see below).

Although not all suspected cases of epilepsy (SCE) identified by Chesnais et al.[10] would have been OAE cases, a study conducted in the same area and at the same time (in 2017–18) ascertained that 93.2% of persons with epilepsy fulfilled the OAE diagnostic criteria[24]. We acknowledge that cysticercosis (another infection associated with epilepsy) has been reported in the Mbam Valley[25]. However, a case-control study conducted in the village of Bilomo (located in the valley) failed to reveal a statistically significant association between seropositivity for cysticercosis and epilepsy, with the authors proposing onchocerciasis as an alternative explanation for the high epilepsy prevalence found in the village[26]. More generally, a better understanding and mapping of the co-endemicity of onchocerciasis and taeniasis/cysticercosis in relation to epilepsy in SSA constitutes an important research gap that needs to be urgently addressed[27].

Currently, EPIONCHO-IBM does not include excess human mortality. A dose-response relationship between mf load and relative risk of mortality has been reported[5]. Bhattacharyya et al., using the ONCHOSIM transmission model, assumed the operation of excess mortality in individuals with OAE, modelled as a reduction of their residual life expectancy[28]. If excess mortality were compensated by high birth rates, as observed in South Sudan (Luís-Jorge Amaral, pers. comm.), incorporation of these processes into the demographic structure of EPIONCHO-IBM would lead to a younger population and likely generate higher OAE incidence rates than those currently predicted. Expanding the age range to 3–18 years for OAE onset[6,8,17] could also be explored, but this was not pursued here, given the lack of mf data beyond 15 years of age in refs. [10,11]. Interestingly, the relative mortality risk as a function of mf load was statistically significantly higher for those aged < 20 years compared to adults[5].

In conclusion, by integrating OAE into EPIONCHO-IBM, via the dose-response relationship between *O. volvulus* mf load in childhood and risk of developing epilepsy later in life reported in ref. [10], the model closely reproduces observed prevalence and incidence after 19 years of annual CDTI in the Mbam Valley area of Cameroon[10,16], as well as the relationship between mf prevalence and epilepsy prevalence across several SSA epidemiological settings[7] (Fig. 4). This suggests that such dose-response relationship captures a fundamental process by which *O. volvulus* infection is associated with epilepsy rather than being an observation specific to the study area, conferring a high degree of generalisability to our modelling framework. Scenario-based simulations highlight the urgency of strengthening MDA-based programmes by increasing treatment frequency and coverage and minimising non-adherence to effect maximal reductions in OAE prevalence and incidence, whose benefits may continue after treatment cessation. In holoendemic settings, alternative and/or complementary strategies will likely be required to markedly reduce transmission and protect gains against OAE. Modelling the impact of larviciding (as done for the Sanaga river[29], of which the Mbam is its most important tributary), and/or of other vector control methods

such as 'slash-and-clear'[30], alone or in combination with a switch to moxidectin MDA would be an important avenue for further work, given the superior efficacy of moxidectin and similar safety profile compared to ivermectin in clinical trials and its potential to accelerate onchocerciasis elimination as shown in modelling studies[31]. Given that our work highlights the contribution to OAE incidence of currently untreated children, moxidectin could help address this gap. Paediatric dose-finding studies seeking to investigate the pharmacokinetics and safety of moxidectin for the treatment of 4–11-year-olds have been completed[32], and modelling studies should be undertaken to investigate the impact of moxidectin MDA on OAE prevalence and incidence compared to ivermectin. Modelling studies should also be undertaken to quantify the impact of supplementing annual CDTI with an extra treatment round targeted at school-age children[33]. More generally, our model can help inform onchocerciasis control and elimination programmes aimed to interrupt transmission as proposed by the World Health Organization 2021–2030 Roadmap on NTDs[34] and consequently eliminate OAE. Our study strengthens the argument to include OAE in future iterations of the GBD Study to fully capture the range of onchocerciasis-associated disease sequelae in addition to the currently considered cutaneous and ocular clinical manifestations[3]. To this end, it will be crucial to incorporate OAE-associated premature mortality into EPIONCHO-IBM.

## Methods

### Epidemiological data

We used the parasitological and epilepsy data as published in ref. [10]. Briefly, in 1991–1993, onchocerciasis surveys were conducted in 25 villages located in the Mbam River Valley, Centre Region of Cameroon. Two skin snips were taken from each consenting participant aged ≥ 5 years with a 2-mm Holth corneoscleral punch from each (right and left) iliac crests and incubated in saline for 24 h, after which emerged microfilariae were counted under a microscope and the arithmetic mean number of microfilariae per skin snip calculated for each individual[10]. Seven of the original 25 villages were selected to be revisited in 2017 (~ 25 years later) for the epilepsy survey. These villages were selected based on the high proportion of the population that had participated in the initial (baseline) 1991–93 parasitological survey and their wide range of community microfilarial load values (18·8–114·5 microfilariae/skin snip). The epilepsy survey used a previously validated and standardised questionnaire to identify suspected cases of epilepsy, SCE)[10]. Of 856 individuals with mean microfilarial (mf) load data when aged 5–10 years in 1991–93, 729 (85%) were located and interviewed in 2017 when aged 30–35 years. A total of 60 individuals out of the 729 (8.2%) were identified as SCE. Their mean mf load (when aged 5–10 years) had ranged from 1 to > 200 microfilariae/skin snip, which was binned into seven categories and for each category, the proportion of SCE was calculated, revealing a dose-response relationship between mf load during childhood and the risk of developing epilepsy later in life[10]. Annual community-directed treatment with ivermectin (CDTI) started in 1998[10], 5–7 years after the baseline parasitological data had been collected. Supplementary Text S2, Supplementary Fig. S5 and Supplementary Tables S4 and S5 provide further details of the Chesnais et al.[10] study that we use in our modelling.

### Incorporation of OAE into EPIONCHO-IBM

We used EPIONCHO-IBM[13]. Briefly, EPIONCHO-IBM is an individual-based, stochastic model tracking the number of adult *O. volvulus* worms of both sexes and microfilariae in the skin within human hosts. Exposure to blackfly bites varies with age and between males and females, in addition to inter-individual exposure heterogeneity as described in Supplementary Text S1, Supplementary Fig. S4 and Supplementary Table S3. We fitted Eq. (1) to the data reproduced in

Supplementary Table S5[10], to parameterise the following relationship between the probability of OAE onset and mf load,

$$Y(M_{(i)}) = \exp\left\{ \beta_1\left[\log\left(M_{(i)}+1\right)\right] + \beta_0 \right\} \qquad (1)$$

where $Y(M_{(i)})$ represents OAE onset probability as a function of individual mf load, $M_{(i)}$ (taken as the mid-point of the binned mf load in Supplementary Table S5), $\beta_0$ is the intercept, such that a mf load of zero can be associated with a non-zero OAE onset probability (to account for sampling error in skin-snipping), and $\beta_1$ is the strength of the association between mf load and the probability of developing OAE. Supplementary Text S3 explains how children in the modelled population become a case of OAE with probability $Y(M_{(i)})$, and provides further details of OAE modelling assumptions. All analyses were undertaken using R, version 4.3.2, with visualisations produced either in base R or using the ggplot2 package, version 3.5.1[35].

### Calibration of EPIONCHO-IBM prior to CDTI
The model was fitted (by least squares) to the observed mf loads in the 5–10-year-olds examined in 1991–93 who were followed up in 2017 (Supplementary Table S4) to estimate the annual biting rate (ABR, no. bites/person/year) that best reproduced the 1991–93 parasitological data. The (endemic equilibrium) modelled mf prevalence (in those aged ≥ 5 years), community microfilarial load (CMFL, geometric mean no. of mf per skin snip (ss) in those aged ≥ 20 years (both sexes)[36]), annual transmission potential (ATP, no. L3/person/year), and infective larval load per fly (no. L3/fly) were compared, by way of validation, with the baseline epidemiological[14] and entomological[15] conditions recorded in the Mbam Valley at the time (Supplementary Text S2). To illustrate uncertainty around the best-fit ABR estimate, the model was also run with the lowest and highest ABR values reported in ref. 15 for the Mbam Valley.

### Sensitivity analyses to pre-intervention conditions
We calculated the overall OAE prevalence for males and females separately and for the total population, varying ABR between 200 and 200,000[13]. We also investigated the relationship between mf prevalence and OAE prevalence when varying the magnitude of the inter-individual exposure heterogeneity parameter in EPIONCHO-IBM, $k_E$, between 0.2 and 0.4[13] (Supplementary Text S1 and Supplementary Table S3). We compared the EPIONCHO-IBM-predicted OAE prevalence with the random-effects logistic meta-regression model fitted by Pion et al.[7] to paired epilepsy–mf prevalence data obtained through their systematic review and meta-analysis of published papers from a broad range of SSA epidemiological settings.

### Modelling the effect of CDTI in the Mbam Valley
In 1998, annual CDTI commenced in all study communities, expanding to the wider area in 2000. Kamga et al.[19] documented levels of therapeutic coverage in the Centre region of Cameroon, where the Mbam Valley is located. These values were used to model the impact of ivermectin treatment until 2017. Between 1998 and 2017, there were 19 rounds of CDTI. We assumed that coverage was 45% from 1998 to 2001, increasing to 65% (of the total population) from 2002[19]. We assumed that children under 5 years were not eligible for treatment and that the proportion of systematic non-adherence (SNA, proportion of eligible population never treated) was 5%. Supplementary Text S1 describes the ivermectin effects on *O. volvulus* included in EPIONCHO-IBM.

Under the CDTI conditions simulated, the predicted epilepsy prevalence in 30–35-year-olds (the ages of those examined in 2017), and the incidence of epilepsy (number of cases per 100,000 person-years) were compared with the values reported in ref. 10. Also, model-predicted epilepsy prevalence values at baseline and 19 years after CDTI were compared with those recorded by Boullé et al.[16] for the

village of Nyamongo (the only village in common with[10,14], and[16]). The mf prevalence of this village was 82.6% prior to the start of CDTI[14]. Epilepsy prevalence for Nyamongo was modelled using the best-fit ABR estimate obtained as described above, as well as with the minimum and maximum ABR values reported in ref. 15 for the study area. Intervention history (duration and coverage) was modelled following[16,19].

### Scenario modelling of the impact of ivermectin treatment on OAE prevalence and incidence
OAE has been reported for transmission conditions ranging from the upper end of mesoendemicity to holoendemicity[6]. We simulated 25 years of CDTI for a) two scenario-based epidemiological settings: hyperendemic (60% mf prevalence at baseline) and holoendemic (80% mf prevalence at baseline); b) two coverage levels: minimal (65% therapeutic coverage of total population, 5% SNA) and enhanced (80% coverage, 1% SNA), and c) two treatment frequencies: annual and biannual (6-monthly). Simulations continued for 75 years after stopping treatment to evaluate the rate of resurgence of OAE prevalence (%) and annual incidence (epilepsy cases/100,000/year). Incidence was calculated as an average every 5 years to smooth variation whilst preserving its overall trend. We also recorded OAE incidence in individuals aged under 5 years (not treated with ivermectin) to quantify their contribution to the overall value and assess how this changed following treatment of the eligible population.

We adhered to the five principles of the Neglected Tropical Diseases (NTD) Modelling Consortium regarding Policy-Relevant Items for Reporting Models in Epidemiology of NTDs (PRIME-NTD), for good practice in NTD modelling[37] (Supplementary Text S4 and Supplementary Table S6).

### Reporting summary
Further information on research design is available in the Nature Portfolio Reporting Summary linked to this article.

## Data availability
All information used for the analyses is contained in the figures, tables, and supplementary information. For this study, EPIONCHO-IBM was parameterised using previously published, publicly available data from Chesnais et al.[10].

## Code availability
The EPIONCHO-IBM model code, as well as the code used for the analyses in this paper, is available at: https://github.com/mrc-ide/EPIONCHO.IBM[38].

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

## Acknowledgements

J.N.S. received funding from the UK Medical Research Council (MRC) 1 + 3 MRes + PhD Studentship in Infectious Disease Epidemiology (MR/S502388/1). M.A.D. and M.G.B. acknowledge funding from the MRC Centre for Global Infectious Disease Analysis (MR/X020258/1), funded by the UK Medical Research Council (MRC). This UK-funded award is carried out in the frame of the Global Health EDCTP3 Joint Undertaking. M.W., M.A.D., and M.G.B. also acknowledge funding by the Bill & Melinda Gates Foundation through the NTD Modelling Consortium (INV-030046). We thank Dr Samit Bhattacharyya (Department of Mathematics, Shiv Nadar University, Uttar Pradesh, India) for valuable discussions in the initial phases of the work. We are also indebted to Prof Charles Newton (Department of Psychiatry, St John's College, University of Oxford, UK) and Luís-Jorge Amaral (Global Health Institute, University

of Antwerp, Belgium) for helpful insights in the more advanced stages of the research. The funders did not have any influence in the study design, analysis, interpretation of results, the writing of the manuscript, or the decision to submit the paper for publication. The corresponding authors had final responsibility for the decision to submit for publication.

## Author contributions

Conceptualisation: J.N.S., J.I.D.H., M.W., and M.G.B. Formal analysis: J.N.S., J.I.D.H., and M.W. Investigation and methodology: J.N.S., J.I.D.H., M.W., and M.G.B. Software: J.I.D.H. and M.A.D. Visualisation: J.N.S., J.I.D.H., M.W., and M.G.B. Supervision: J.I.D.H., M.W., and M.G.B. Writing the original draft: J.N.S. and M.G.B. Writing: review and editing: J.N.S., J.I.D.H., M.W., M.A.D., R.C., and M.G.B.

## Competing interests

The authors declare no competing interests.
