## [Peer Review File · Nature Communications]

Modelling onchocerciasis-associated epilepsy and the impact of ivermectin treatment on its prevalence and incidenceREVIEWER COMMENTS

Reviewer #1 (Remarks to the Author):

The work you have undertaken to model the relationship between onchocerciasis prevalence and epilepsy represents a significant step forward in understanding how interventions might impact the prevalence and incidence of epilepsy in onchocerciasis-endemic regions. I have several questions concerning calibration and threshold estimations:

1. Calibration:

- Could you elaborate on the choice of using a specific ABR value for calibration? What were the key considerations in selecting this figure, and how do you ensure it adequately represents the uncertainty in the ecological setting, even for the single site studied?

2. Sensitivity and Uncertainty Analysis:

- It would be beneficial to understand any sensitivity analyses that were conducted. How does uncertainty in ABR influence your results, specifically in estimating mf prevalence, and what does this imply for the generalizability of your findings across different settings? Currently, OAE is shown as a function of ABR (Figure 3). However, an error bar around mf prevalence in Figure 2 would be helpful for examining model-predicted uncertainty, primarily due to uncertainty in ABR for the study site.

3. Threshold mf and OAE:

- The methodology used for the dose-response relationship between mf loads in skin snips and epilepsy is crucial. Were your models able to identify mf thresholds below which epilepsy is not possible or eliminated?

With these questions addressed, I recommend publication.

Reviewer #1 (Remarks on code availability):

The code provides a readme file providing essential information on installing and running the code. I recommend also to create readme file with instructions on the structure of the code as to how it works algorithmically to make it transparent and possibly extendable.

Reviewer #2 (Remarks to the Author):

In this manuscript the authors present an IBM investigating the relationship between Onchocerciasis infection and subsequent onset of epilepsy. The authors make updates to their stochastic model to simulate this relationship and calibrate it to epidemiological data. They investigate the impact of ABR, exposure heterogeneity, and the impact of treatment on OAE occurrence in different endemicities.

Whilst I appreciate the thoroughness with which the model fit has been explored, I think there are too many figures - some of the fitting information can go in the SM.

The methods state that there are 60 individuals with SCE but there are only 7 data points on Figure 1. Where are the rest of them?

The authors have made the assumption that the probability of epilepsy is tied to the density of mf in a skin snip, but have provided little evidence in the introduction to support this. Reading to the bottom of the discussion, I appreciate that this is listed as a limitation but I would expect there to be a little more evidence around this and more coverage to set the scene in the intro.

I am interested whether the reduction shown in figure 2 is actually because of treatment. The reduction is only $\sim 1.5\%$ and the errors overlap.

The increased probability that males develop OAE, is this an artefact of greater exposure in men

than women? Is the relationship standardised first?

The assumption being in figure 5 that everyone in those 100,000 is similarly exposed and has the same probability of developing OAE? I feel like this is a bit of a stretch and a misleading figure to show without more upfront commentary on this point. Again I appreciate it is listed as a limitation but it's quite a significant simplification, which if considering the figure has quite strong ramifications for treatment.

Reviewer #2 (Remarks on code availability):

The readMe file has a figure of temperature~pressure (not very informative). Everything else looks fine to me. I have not checked the nitty gritty of the code but it everything is accessible.

Reviewer #3 (Remarks to the Author):

The ms aims to provide a computer model for onchocerciasis – associated epilepsy and to predict the impact of ivermectin treatment on its prevalence and incidence. The group uses already published data from the Mbam valley in Cameroon collected in 1991-93 (onchocerciasis survey data and OAE retrospective data) and 2017 (onchocerciasis and OAE data). The reviewer is no modeling expert but the modeling process appears to be straight forward. However, the ms has two major weaknesses: (1) Sample collection criteria at baseline and at follow up are poorly described. Thoughtful selection of study villages and study participants is not only important for onchocerciasis surveys but especially for assessing the prevalence/incidence of OAE. Self- selection of OAE affected individuals or hiding conditions because of stigma can be important confounders. The best model is only as strong as the data that go into the model and the assumptions made in the model. (2) The authors give the impression that the model applies for OAE in all areas in 'SSA' endemic for onchocerciasis. Epilepsy that is geographically associated with the distribution of onchocerciasis was first described in East Africa (Uganda) and was thought to be limited to *Simulium neavei* transmitted *O. volvulus*. Much later OAE was also described from other areas in Central and West Africa. Therefore, the prevalence and intensity of *O. volvulus* infection are most likely not the only predictors for OAE. It is possible that the described model applies only to the small but well studied area in central Cameroon.

Members of the group have already published a large number of studies about the geographical association of epilepsy and onchocerciasis. They have also described that epilepsy recedes upon intensification of onchocerciasis intervention. The main message of the present paper is probably correct and may help the onchocerciasis elimination program of WHO and partner countries. However, the detailed objective of the present study is somewhat unclear. For what purpose should the described model be used? Does the model also apply to other endemic areas? What is next for this model?

Selected specific comments:

Title. The title gives the impression that a general model for OAE was developed which is not true. At least the country needs to be added.

Line 16. Donated ivermectin can be used in children 5 years and older, and moxidectin has been evaluated for children 4 years and older. To call for expensive treatment studies in pre-school aged children (aged 3-4) requires a very strong scientific justification. Since these young children are rarely mf positive, significant reduction of the mf reservoir in older subjects may be sufficient to reduce the risk of early development of OAE.

Line 23, 27, 28. Is it necessary to introduce abbreviations like SSA, OCP, or APOC that are only used once or twice in the entire paper?

Line 28. While previous control programs were mentioned, it would be important to mention the WHO program for the elimination of human onchocerciasis (and potentially OAE). This program is in agreement with the roadmap for neglected tropical diseases of the WHO, is mostly based on mass drug administration and has to ultimate goal of elimination of *O. volvulus* transmission.

Line 56-64. Although the authors mostly refer to already published data, important background information is missing that should be included in the EPIONCHO model. For example, high rates of

epilepsy caused by cysticercosis have been described in parts of Cameroon (Angwafor et al. 2021). How was OAE and cysticercosis associated epilepsy differentiated? What proportion of epilepsy cases was ascribed to which worm infection in the model? If the authors don't have data on other infections, they should at least mention this additional limitation in their limitation section.

Line 117. How was 'holoendemic' endemicity (80% mf rate) calculated? Children under the age of 3 years are usually mf negative and mf rates in preschool and primary school aged children are usually very low. Migrants from larger cities such as Yaounde are also mostly mf negative.

Therefore, it is difficult to achieve a sex/age standardized mf rate of at least 80%.

Line 139. The table should not only include the publication year, but the year when data were collected and the exact villages where data were collected.

Line 292 The model predicts that 3-4 year old children contribute 10-12% of the total OAE. What was the mf rate in this age group? Or at least what was the exposure to Simulium bites? Children in this age group are usually not working or playing close to the rivers with high annual biting rates.

Reviewer #3 (Remarks on code availability):

Although I worked together with modellers before, I am not a computer modeler and I did not try to install and run the code.

Responses to Reviewers' Comments (NCOMMS-24-16977)

REVIEWER COMMENTS

Reviewer #1 (Remarks to the Author):

1.1. The work you have undertaken to model the relationship between onchocerciasis prevalence and epilepsy represents a significant step forward in understanding how interventions might impact the prevalence and incidence of epilepsy in onchocerciasis-endemic regions.

Response by authors to 1.1

We thank the reviewer for these appreciative comments and more generally for the valuable insights offered.

I have several questions concerning calibration and threshold estimations:

1.2. Calibration:

Could you elaborate on the choice of using a specific ABR value for calibration? What were the key considerations in selecting this figure, and how do you ensure it adequately represents the uncertainty in the ecological setting, even for the single site studied?

Response by authors to 1.2

We thank the referee for raising this important question. We fitted our model, using least squares, to the means of age-specific microfilarial load data for those aged 5-10 years in 1991-1993 who were followed up for epilepsy 25 years later ($n = 729$) as described in Chesnais et al. 2018, Appendix 6: Parasitological results according to the age. We have presented a summary of our fitting in Table S5: Comparison of measures of central tendency and variation for microfilarial (mf) load recorded by Chesnais et al. and generated by EPIONCHO-IBM (annual biting rate (ABR) = 41,992 bites/person/year, $k_E = 0.3$). The ABR found to best reproduce these means was 41,922. To validate this, we compared parasitological and entomological outputs from our model when running it with an ABR of 41,922, to the data presented by Pion et al. 2004 and Barbazan et al. 1998 (who collected these data in 1991-1993 and 1993-1994, respectively). We show these results in Table 1: Comparison Between Observed and Modelled Baseline Epidemiological and Transmission Conditions. EPIONCHO-IBM closely reproduces the baseline microfilarial prevalence, community microfilarial load, annual transmission potential and number of L3 larvae per fly as reported in these studies, which were conducted in the

same region and approximately at the same time as the parasitological data presented in Chesnais et al. 2018 (prior to control interventions). For further validation, when again using an ABR of 41,922 and this time simulating the 19 years of intervention history and coverage levels documented in the region (Kamga et al. 2017), our model closely reproduced the epilepsy prevalence and incidence reported by Chesnais et al. 25 years later, i.e. in the 30-35-year age group (observed prevalence = 8.2%, modelled prevalence = 7.6%; observed incidence = 350, modelled incidence = 317 cases/100,000 person-years). See also Table S6: Incidences of onchocerciasis-associated epilepsy (OAE) generated by EPIONCHO-IBM compared to values in Chesnais et al. 2018.

However, we thank the reviewer for highlighting the importance of representing uncertainty in ecological setting / transmission conditions in the study area. In fact, the study had been conducted in several villages located close to the Mbam river valley (see Figure S1, where we reproduce the map of the Chesnais et al. 2018 study location). To capture this uncertainty, we have now run our model using the smallest and largest values of the range of ABRs documented in the region at baseline by Barbazan et al. 1998 (~14,000 and ~100,000 bites/person/year, respectively). In the new Figure S3, we have plotted the mean age-specific microfilarial loads of those aged 5-10 years in 1991-1993 from the Chesnais et al. study and their associated 95% confidence intervals (data points and error bars). We have also plotted the mean microfilarial loads generated by the model for ABR = 41,922 (grey line) and those corresponding to ABR = 14,000 and ABR = 100,000, encompassing a shaded area to illustrate uncertainty. Our modelled microfilarial loads for ABR of 41,922 closely match the datapoints, and the shaded area generated between ABR of 14,000 and 100,000 largely falls within the 95% confidence intervals of the data.

To further clarify that the parasitological and entomological data of Pion et al. and Barbazan et al. were collected at baseline during approximately the same period as the Chesnais et al. data, we have added a footnote to Table 1 (also see Response 3.11).

We have modified the text in the Methods as follows:

“The model was fitted (by least squares) to the observed mf loads in the 5–10-year olds examined in 1991–93 who were followed-up in 2017 (Supplementary Table S1) to estimate the annual biting rate (ABR, no. bites/person/year) that best reproduced the 1991–93 parasitological data. The (endemic equilibrium) modelled mf prevalence (in those aged ≥ 5 years), community microfilarial load (CMFL, geometric mean no. of mf per skin snip (ss) in those aged ≥ 20 years (both sexes) [14]), annual transmission potential (ATP, no. L3/person/year), and infective larval load per fly (no. L3/fly) were compared, by way of validation, with the baseline epidemiological [15] and entomological [16] conditions recorded in the Mbam Valley at the time (Supplementary Text S1). To illustrate

uncertainty around the best ABR estimate, the model was also run with the lowest and highest ABR values reported in [16] for the Mbam valley.”

And in the Results:

“An estimated ABR = 41,922 bites/person/year was found to be the best biting rate estimate capable of generating mf loads consistent with those recorded in 1991–93 in [10]. Supplementary Table S5 presents measures of central tendency and variation in mf loads for children aged 5–10 years for the 729 individuals who were examined in 1991–93 and followed up for epilepsy in 2017 compared with the same metrics derived from EPIONCHO-IBM.

Table 1 compares the epidemiological and entomological results recorded in the Mbam Valley at baseline [15,16], with EPIONCHO-IBM-generated values for the model calibrated with ABR = 41,922, indicating that model outputs are consistent with observations. The ranges around the modelled values are those corresponding to ABR of 14,000 and 100,000 by way of representing uncertainty in the transmission conditions in the study area [16].

Figure S3 presents the observed age-specific mf loads for children aged 5–10 years in 1991–93, their 95% confidence intervals (95% CI), the modelled mf loads for ABR = 41,922 and the corresponding results for ABR = 14,000 and ABR = 100,000.”

New Figure S3 has been included in the revised Supplementary Information.

1.3. Sensitivity and Uncertainty Analysis:

It would be beneficial to understand any sensitivity analyses that were conducted. How does uncertainty in ABR influence your results, specifically in estimating mf prevalence, and what does this imply for the generalizability of your findings across different settings? Currently, OAE is shown as a function of ABR (Figure 3). However, an error bar around mf prevalence in Figure 2 would be helpful for examining model-predicted uncertainty, primarily due to uncertainty in ABR for the study site.

Response by authors to 1.3

We thank the referee for the opportunity to clarify this and for raising an important point, which we address in the revised version of the MS. In EPIONCHO-IBM, baseline mf prevalence in the community for any given (from hypo- to holoendemic) setting is determined by the ABR as shown in the paper describing the model (Hamley et al., 2019 Figure 2: Pre-intervention *Onchocerca volvulus* microfilarial prevalence [...] versus the annual biting rate of simuliid vectors).

Our Figure 3 already shows a sensitivity analysis of the modelled OAE prevalence, by varying the ABR between 200 and 200,000 bites/person/year. In brief, we varied the ABR between these values and obtained the corresponding mf prevalence which links to the corresponding OAE prevalence by the mechanistic relationship (between mf load and onset probability of developing epilepsy) incorporated into EPIONCHO-IBM.

Also, a further sensitivity analysis is given in Figure 4, where we vary mf prevalence (by varying ABR) and exposure heterogeneity (by varying the k_E parameter) to generate OAE prevalence. The generalisability of our results for Africa is highlighted by the fact that our transmission model-derived relationships (coloured lines) match remarkably closely the meta-regression relationship (black line) reported by Pion et al. 2009. These authors had conducted a systematic review and meta-analysis of published studies across sub-Saharan Africa. In order to bring this point to the fore, we have now moved previous Figure S3 to the main text Figure 4 panel B. This figure provides the data (and error bars for both mf prevalence and epilepsy prevalence) used by Pion et al. 2009, together with our modelled relationships. We have modified text in the Introduction to highlight this important study:

“Epidemiological studies have indicated an association between onchocerciasis and epilepsy [6–8]. In particular, the systematic review and meta-analysis presented by Pion et al. [7] evaluated the relationship between onchocerciasis prevalence and that of epilepsy at the community level for studies distributed from West to East Africa [7]. Onchocerciasis-associated epilepsy (OAE) mainly occurs in highly endemic regions with intense ongoing transmission.”

Figure 2 plots epilepsy prevalence with error bars for the point estimates given in Boullé et al. 2019 for 1992 and 2017, not mf prevalence. However, we acknowledge that we were possibly unclear owing to our efforts to provide additional detail about the baseline mf prevalence for Nyamongo. In the original submission, we had used $ABR = 41,922$ to model epilepsy prevalence at baseline and following 19 years of ivermectin treatment according to the intervention history reported by Boullé et al. and by Kamga et al. 2017. However, as suggested by the reviewer, we have now included the variation in epilepsy prevalence that derives from using ABR 14,000 and 100,000 (as explained above in Response to 1.2). New Figure 2 demonstrates that by incorporating this uncertainty, our model generates outcomes that fall within the 95% CIs around the epilepsy prevalence reported by Boullé et al. for 1992 and 2017.

We have clarified these points in the Methods as follows:

“Also, model-predicted epilepsy prevalence values at baseline and 19 years after CDTI were compared with those recorded by Boullé et al. [18] for the village of Nyamongo (the only village in common with [10], [15] and [18]). The mf prevalence of this village

was 82.6% prior to the start of CDTI [15]. Epilepsy prevalence for Nyamongo was modelled using the best ABR estimate obtained as described above, as well as with the minimum and maximum ABR values reported in [16] for the study area. Intervention history (duration and coverage) was modelled following [17,18]”.

New Figure 2 legend now reads:

“Figure 2. Observed and EPIONCHO-IBM-predicted prevalence of epilepsy in Nyamongo, Mbam Valley, Cameroon at baseline and after 19 years of annual community-directed treatment with ivermectin (CDTI). The squares are the data from Boullé et al. [18] and the error bars are the 95% confidence intervals (95% CIs) around the data. The epilepsy prevalence in 1991–92 was 6.4% (95% CI = 4.7–8.6%) and in 2017 it was 5.1% (95% CI = 4.0–6.6%). The grey line with black circles represents the OAE prevalence trend predicted by EPIONCHO-IBM for annual biting rate (ABR) = 41,922 bites/person/year; the grey shaded area indicates uncertainty around OAE prevalence when simulating a minimum ABR = 14,000 and a maximum of 100,000 [16]. The solid black vertical line indicates the start of annual CDTI at 45% coverage; the vertical dashed line indicates the increase in coverage to 65% (of total population). Therapeutic coverage levels are informed by Kamga et al. [17].

New Figure 4 legend now reads:

Figure 4. Onchocerciasis-associated epilepsy (OAE) prevalence as a function of *Onchocerca volvulus* microfilarial (mf) prevalence for three values of inter-individual exposure heterogeneity. The coloured (beaded) lines correspond to the EPIONCHO-IBM-predicted OAE prevalence. **(A)** The solid black line and shaded grey area represent, respectively, the logistic meta-regression model fitted by Pion et al. (2009) to paired epilepsy and mf prevalence data from African settings and associated 95% confidence interval [7]. **(B)** The markers represent data for 7 countries across sub-Saharan Africa to which the meta-regression model in [7] was fitted (CAR, Central African Republic; Uganda, two studies: Kaiser et al. and Kipp et al. [7]). The starting point of mf prevalence for each coloured line corresponds to the modelled threshold biting rate (minimum annual biting rate for onchocerciasis endemicity) for each value of the k_E parameter (~75; ~180 and ~350 bites/person/year, respectively, for $k_E = 0.2, 0.3$ and 0.4).

1.4. Threshold mf and OAE:

The methodology used for the dose-response relationship between mf loads in skin snips and epilepsy is crucial. Were your models able to identify mf thresholds below which epilepsy is not possible or eliminated?

Response by authors to 1.4

In Figure 1 we show the modelled relationship between mf load (in childhood) and OAE onset probability based on the binned mf loads reported in Chesnais et al. 2018. Because of potential sampling error inherent in skin snipping, we have assumed that even for 0 mf per skin snip there could be an associated probability of epilepsy onset. Therefore, at an individual level, our dose-response relationship does not have a cut-off threshold neither can one be inferred from the (binned) data from Chesnais et al. we have used.

At the population level, it would be possible to have epilepsy for any level of onchocerciasis endemicity, as per Figure 4B (see response to 1.3). As an example, we have run, for the benefit of the reviewer, a scenario-based simulation using a mesoendemic setting (40% mf prevalence and ABR of 500). The resulting baseline OAE prevalence was 1.25% (all ages). Under this scenario, the incidence of epilepsy cases after 25 years of annual CDTI would decline to 0 per 100,000 persons per year (i.e. would be eliminated). We have not included this in the revised manuscript because, by and large, OAE has mainly been documented in regions with higher levels of onchocerciasis endemicity.

With these questions addressed, I recommend publication.

We thank the reviewer for their confidence in our work; their valuable contributions have stimulated a productive discussion and we hope we have addressed all the queries satisfactorily.

Reviewer #1 (Remarks on code availability):

The code provides a readme file providing essential information on installing and running the code. I recommend also to create readme file with instructions on the structure of the code as to how it works algorithmically to make it transparent and possibly extendable.

We thank the reviewer for helping to enhance the accessibility and clarity of our code for other users. We refer the reviewer to the flowchart of the process overview included in the EPIONCHO-IBM Readme file for the base model found at: <https://github.com/mrc-ide/EPIONCHO.IBM?tab=readme-ov-file>. We have added a further flowchart of process overview which specifically describes how the OAE module works within the EPIONCHO-IBM model on the same Readme file. Note that the OAE module can be added or removed from the base EPIONCHO-IBM package.

Reviewer #2 (Remarks to the Author):

2.1. In this manuscript the authors present an IBM investigating the relationship between Onchocerciasis infection and subsequent onset of epilepsy. The authors make updates to their stochastic model to simulate this relationship and calibrate it to epidemiological data. They investigate the impact of ABR, exposure heterogeneity, and the impact of treatment on OAE occurrence in different endemicities.

Response by authors to 2.1

We appreciate the reviewer's comments on the aim of our work.

2.2. Whilst I appreciate the thoroughness with which the model fit has been explored, I think there are too many figures - some of the fitting information can go in the SM.

Response by authors to 2.2

We thank the reviewer for their positive comment regarding the thoroughness of our model fitting work. However, we would like to maintain the number of figures in the main text as we also need to address the queries of the other referees, and in particular those referring to the generalisability of our results (see Response to 1.3)

2.3. The methods state that there are 60 individuals with SCE but there are only 7 data points on Figure 1. Where are the rest of them?

Response by authors to 2.3

In Figure 1 we plot the 7 points related to the 7 bins of mf load reported by Chesnais et al. 2018. We refer the reviewer to Table S2, where the 60 suspected cases of epilepsy (SCE) individuals are assigned to mf load bins which are 7 in number (i.e. 0, 1 – 5, 6 – 20, 21 – 50, 51 – 100, 101 – 200 and >200 microfilariae per skin snip. We appreciate the opportunity to provide further clarification, and therefore we have added information to the legend of Figure 1, which now reads:

"Figure 1. Probability of OAE onset as a function of microfilarial load. Calculated OAE onset probabilities were plotted against the midpoints of the seven binned (mean) mf loads from Chesnais et al. [10], reproduced in Table S2 [...]."

For further clarification we have added the following footnote to Table S2: "**For each participant 2 skin snips were taken (each from the right and left iliac crests) and incubated for 24 h in saline to enumerate the emerged microfilariae. The arithmetic

mean number of microfilariae per skin snip for each individual was calculated. These means were binned into 7 mf load categories.”

2.4. The authors have made the assumption that the probability of epilepsy is tied to the density of mf in a skin snip, but have provided little evidence in the introduction to support this. Reading to the bottom of the discussion, I appreciate that this is listed as a limitation but I would expect there to be a little more evidence around this and more coverage to set the scene in the intro.

Response by authors to 2.4

This is an important point that we wish to clarify. Our modelling is predicated on the empirical dose-response relationship, documented by Chesnais et al. 2018, between microfilarial density in childhood and the risk of developing epilepsy later in life. Therefore, our modelled relationship between mf load and epilepsy onset probability (Figure 1) is not an ‘assumption’ but based on real-world data. If anything, the fact that we have used this data-driven relationship to incorporate epilepsy as an output of our model, and that our model-derived prevalence and incidence results are remarkably close to those reported by Chesnais et al. for the individuals participating in the epilepsy questionnaire 25 years later, lends support to the operation of such dose-response relationship (see also new Figure 2, new Figure 4 and Response to 1.3).

The reviewer suggests that additional information be provided in the Introduction regarding the relationship between the probability of developing epilepsy and the density of mf in a skin snip. We refer the reviewer to the following paragraph that had already been included in our original submission:

“The retrospective cohort studies of Chesnais et al. [10,11] provide the strongest evidence of temporality in the relationship between past *O. volvulus* microfilarial (mf) load and epilepsy incidence. Villagers who, aged between 5–10 years [10] or 5–15 years [11] had been assessed for mf load, were re-visited 25–30 years later to investigate the incidence of epilepsy in the intervening period, indicating a dose-response relationship between mf load in childhood and the risk of developing epilepsy later in life.”

However, we do appreciate the need to include more detail in the main text regarding how Chesnais et al. 2018 conducted their study and reached their conclusions. We have revised the text in Methods as follows:

“Epidemiological Data

We used the parasitological and epilepsy data as published in [10]. Briefly, in 1991–1993 onchocerciasis surveys had been conducted in 25 villages located in the Mbam river

valley, Centre Region of Cameroon. Two skin snips were taken from each consenting participant aged ≥ 5 years with a 2-mm Holth corneoscleral punch from each (right and left) iliac crests and incubated in saline for 24 h, after which emerged microfilariae were counted under a microscope and the arithmetic mean number of microfilariae per skin snip calculated for each individual [10]. Seven of the original 25 villages were selected to be revisited in 2017 (approximately 25 years later) for the epilepsy survey. These villages were selected based on the high proportion of the population that had participated in the initial (baseline) 1991–93 parasitological survey and their wide range of community microfilarial load values (18.8–114.5 microfilariae/skin snip). The epilepsy survey used a previously validated and standardised questionnaire to identify suspected cases of epilepsy, SCE) [10]. Of 856 individuals with mean microfilarial (mf) load data when aged 5–10 years in 1991–93, 729 (85%) were located and interviewed in 2017, when aged 30–35 years. A total of 60 individuals out of the 729 (8.2%) were identified as SCE. Their mean mf load (when aged 5–10 years) had ranged from 1 to >200 microfilariae/skin snip, which was binned into seven categories and for each category, the proportion of SCE was calculated, revealing a dose-response relationship between mf load during childhood and the risk of developing epilepsy later in life [10]. Annual community-directed treatment with ivermectin (CDTI) started in 1998 [10], 5–7 years after the baseline parasitological data had been collected. Supplementary Information Text S1, Figure S1 and Tables S1–S2 provide further details of the Chesnais et al. [10] study that we use in our modelling.”

2.5. I am interested whether the reduction shown in figure 2 is actually because of treatment. The reduction is only $\sim 1.5\%$ and the errors overlap.

Response by authors to 2.5

We refer the reviewer to the Results section “Effect on OAE prevalence and incidence of 19 years of annual CDTI” where we have reported the proportional reduction in OAE prevalence for the Nyamongo village after 19 years of treatment illustrated in Figure 2. According to Boullé et al. 2019, there was a 20.3% reduction in epilepsy prevalence, calculated as $100 - (\text{prevalence in 2017}/\text{prevalence in 1992})$, with our modelling yielding a reduction of 19.7%. The reviewer is right that the error bars around the data points provided by Boullé et al. are wide, owing to small sample sizes. To better capture heterogeneity in transmission conditions, as suggested by Referee 1, we have re-run the model with different annual biting rate values under the same history of intervention. New Figure 2 indicates that our modelling results fall within the 95% CIs around the data points (see also Response to 1.3). In addition, when we run the scenario-based simulations for a holoendemic setting (the mf prevalence of Nyamongo was 83% at baseline), our modelled proportional reduction in epilepsy prevalence after 19 years of

annual treatment at a coverage of 65% is 20.6% (Figure 5), supporting that the reduction reported by Boullé et al. is because of treatment.

2.6. The increased probability that males develop OAE, is this an artefact of greater exposure in men than women? Is the relationship standardised first?

Response by authors to 2.6

In the Results section we have stated that the prevalence (and 95% CI) of epilepsy reported in [10] for those aged 30–35 years (after 19 rounds of annual CDTI) was 9.5% (7.0–13.0%) in males and 6.8% (4.6–9.9%) in females, compared to our model-generated prevalence of 8.6% in males and 6.7% in females. Chesnais et al. 2018 found that the difference between males and females was not statistically significant. In EPIONCHO-IBM, for the 3–15-year age range (our modelled age range for epilepsy onset), the exposure function shown in Figure S2 (estimated by fitting an age- and sex-structured model to data from Cameroon by Filipe et al. 2005) indicates that boys have a higher exposure than girls. There is no sex-specific susceptibility to developing epilepsy in our model, so the referee is correct in suggesting that the difference in epilepsy prevalence between the sexes (Figure 3) is likely due to exposure.

We have added the following text to the Discussion to address this point:

“The higher epilepsy prevalence for males compared to females in Figure 3 results from the age- and sex-exposure functions in EPIONCHO-IBM (Supplementary Information Figure S2). For the 3–15-year age group (contributing to OAE onset in our model), boys are more highly exposed than girls, with the exposure of both being greater than 0 at birth. This enables the model to capture the high mf loads reported in [10] for children aged 5–10 years (Tables S1 and S5) and to generate a modelled epilepsy prevalence in the 30–35-year age group (the one participating in the epilepsy survey conducted in [10], 25 years after collection of parasitological data in 1991–93) of 8.6% in males and 6.7% in females (compared to the observed 9.5% in males and 6.8% in females) after accounting, in the model, for the 19 years of CDTI experienced in the study area. According to [10], the difference in epilepsy prevalence between the sexes was not statistically significant, and in our model we do not assume a differential susceptibility to developing OAE according to sex; therefore, our results are due to EPIONCHO-IBM exposure functions. These functions had been derived from fitting an age- and sex-structured precursor of our model to data on mf infection profiles from Cameroon, as described in Supplementary Information Text S2.”

2.7. The assumption being in figure 5 that everyone in those 100,000 is similarly exposed and has the same probability of developing OAE? I feel like this is a bit of a stretch and a misleading figure to show without more upfront commentary on this point. Again I appreciate it is listed as a limitation but it's quite a significant simplification, which if considering the figure has quite strong ramifications for treatment.

Response by authors to 2.7

Figure 5 provides the mean dynamics for OAE prevalence (panel A) and incidence (panel B) from 400 model runs for a modelled population of 400 individuals. It does not mean that 100,000 individuals are similarly exposed or have the same probability of developing OAE. The probability of developing OAE is determined by the relationship between mf load in childhood and the onset epilepsy probability presented in Figure 1, which, as we have clarified in Response to 2.2 is based on the dose-response relationship reported by Chesnais et al. 2018. In EPIONCHO-IBM, individuals are exposed according to their age and sex (Figure S2 of Supplementary Information) and their level of individual exposure is drawn from a gamma distribution with parameter k_E (Text S2 and Table S3). The combination of age-, sex- and individual-specific exposure determines their acquisition of infection and ultimately their mf load, which we can assess in the age group of interest (3–15 years) for subsequent development of epilepsy. We then use the model to calculate the prevalence and incidence of epilepsy in the population and express the former as a percent and the latter per 100,000 persons per year (as used in other work reporting on epilepsy). We appreciate the reviewer highlighting our need for further clarification on how the model accounts for exposure heterogeneity and have included text to this effect in the Methods.

“We used EPIONCHO-IBM [13]. Briefly, EPIONCHO-IBM is an individual-based model tracking the number of adult *O. volvulus* worms of both sexes and microfilariae in the skin within human hosts. Exposure to blackfly bites varies with age and between males and females, in addition to inter-individual exposure heterogeneity as described in Text S2, Figure S2 and Table S3 of Supplementary Information.”

Reviewer #2 (Remarks on code availability):

The readMe file has a figure of temperature~pressure (not very informative). Everything else looks fine to me. I have not checked the nitty gritty of the code but it everything is accessible.

We are pleased the reviewer found our code accessible. A temperature – pressure relationship had erroneously been included in the Readme file which has now been deleted. We thank the reviewer for pointing this out.

Reviewer #3 (Remarks to the Author):

3.1. The ms aims to provide a computer model for onchocerciasis – associated epilepsy and to predict the impact of ivermectin treatment on its prevalence and incidence. The group uses already published data from the Mbam valley in Cameroon collected in 1991-93 (onchocerciasis survey data and OAE retrospective data) and 2017 (onchocerciasis and OAE data). The reviewer is no modeling expert but the modeling process appears to be straight forward.

Response by authors to 3.1

We thank the reviewer for stating that our modelling process is understandable for a non-modeller. We would like to clarify that the 1991–93 onchocerciasis surveys focused on the parasitological data but no epilepsy data were collected, whereas in 2017 epilepsy was assessed but parasitological data were not collected, according to Chesnais et al. 2018.

3.2. However, the ms has two major weaknesses: (1) Sample collection criteria at baseline and at follow up are poorly described. Thoughtful selection of study villages and study participants is not only important for onchocerciasis surveys but especially for assessing the prevalence/incidence of OAE. Self- selection of OAE affected individuals or hiding conditions because of stigma can be important confounders. The best model is only as strong as the data that go into the model and the assumptions made in the model.

Response by authors to 3.2

We acknowledge that we had not provided sufficient detail describing how the primary data had been collected by Chesnais and co-workers (2018) in the original submission. In response to this query as well as to Referee 2 (2.4), we have added the following to the Methods section under Epidemiological data:

“We used the parasitological and epilepsy data as published in [10]. Briefly, in 1991–1993 onchocerciasis surveys had been conducted in 25 villages located in the Mbam river valley, Centre Region of Cameroon. Two skin snips were taken from each consenting participant aged ≥ 5 years with a 2-mm Holth corneoscleral punch from each (right and left) iliac crests and incubated in saline for 24 h, after which emerged microfilariae were

counted under a microscope and the arithmetic mean number of microfilariae per skin snip calculated for each individual [10]. Seven of the original 25 villages were selected to be revisited in 2017 (approximately 25 years later) for the epilepsy survey. These villages were selected based on the high proportion of the population that had participated in the initial (baseline) 1991–93 parasitological survey and their wide range of community microfilarial load values (18.8–114.5 microfilariae/skin snip). The epilepsy survey used a previously validated and standardised questionnaire to identify suspected cases of epilepsy, SCE) [10]. Of 856 individuals with mean microfilarial (mf) load data when aged 5–10 years in 1991–93, 729 (85%) were located and interviewed in 2017, when aged 30–35 years. A total of 60 individuals out of the 729 (8.2%) were identified as SCE. Their mean mf load (when aged 5–10 years) had ranged from 1 to >200 microfilariae/skin snip, which was binned into seven categories and for each category, the proportion of SCE was calculated, revealing a dose-response relationship between mf load during childhood and the risk of developing epilepsy later in life [10]. Annual community-directed treatment with ivermectin (CDTI) started in 1998 [10], 5–7 years after the baseline parasitological data had been collected. Supplementary Information Text S1, Figure S1 and Tables S1–S2 provide further details of the Chesnais et al. [10] study that we use in our modelling.”

The questionnaire used to interview the participants (or their family members in the case that individuals were absent or dead) during the 2017 survey had been developed by the Institut d'Épidémiologie Neurologique et de Neurologie Tropicale (Limoges, France). Therefore, we are confident that the primary data were carefully collected and the individuals participating in the epilepsy survey thoughtfully selected.

3.3. The authors give the impression that the model applies for OAE in all areas in 'SSA' endemic for onchocerciasis. Epilepsy that is geographically associated with the distribution of onchocerciasis was first described in East Africa (Uganda) and was thought to be limited to *Simulium neavei*-transmitted *O. volvulus*. Much later OAE was also described from other areas in Central and West Africa. Therefore, the prevalence and intensity of *O. volvulus* infection are most likely not the only predictors for OAE. It is possible that the described model applies only to the small but well studied area in central Cameroon.

Response by authors to 3.3

Although it is true that our model uses well-characterised data on *O. volvulus* mf load and epilepsy derived from the study in the Mbam valley of Cameroon, our results suggest that the dose-response relationship reported by Chesnais et al. 2018 is fundamental to the process by which onchocerciasis is associated with epilepsy more

generally. By parameterising this relationship and including it in the model, we are essentially testing *in silico* the hypothesis that the greater the microfilarial load of an individual during childhood, the greater the probability of developing epilepsy later in life. Our model is by design parsimonious, yet: i) it captures the prevalence and incidence in the study area 25 years later after simulating the intervention history; ii) it reproduces the proportional reduction in epilepsy prevalence after 19 years of CDTI in the village of Nyamongo (Figure 2), and iii) it provides a relationship between microfilarial prevalence and epilepsy prevalence that is consistent with the patterns across sub-Saharan Africa reported by Pion et al. 2009 (new Figure 4 and Response to 1.3). See also Response 3.5, where we describe additional text included in the Discussion to address this point.

3.4. Members of the group have already published a large number of studies about the geographical association of epilepsy and onchocerciasis. They have also described that epilepsy recedes upon intensification of onchocerciasis intervention. The main message of the present paper is probably correct and may help the onchocerciasis elimination program of WHO and partner countries. However, the detailed objective of the present study is somewhat unclear.

For what purpose should the described model be used? Does the model also apply to other endemic areas? What is next for this model?

Response by authors to 3.4

Purpose: We refer the reviewer to the last paragraph of the Introduction in the revised submission, where we have better clarified the objectives of our study to address the point made by the referee:

“There is renewed interest in understanding the consequences of large-scale onchocerciasis control and elimination programmes on disease burden in SSA, expanding the range of sequelae beyond ocular and skin disease thus far included in the GBD Study [3]. Therefore, we incorporated OAE into our stochastic EPIONCHO-IBM transmission model [13] to: i) test *in silico* the dose-response relationship between *O.volvulus* mf load in childhood and the probability of developing epilepsy later in life as crucial to the process by which onchocerciasis is associated with epilepsy; ii) validate the model by testing its ability to capture the results presented in [10] as well as more broadly in [7], and iii) investigate the effect of long-term ivermectin MDA on predicted prevalence and incidence of epilepsy under a range of epidemiological and programmatic scenarios.”

Applicability to other endemic areas: We have already addressed this in Response to 3.3.and 1.3.

The described model can be used to obtain values of OAE prevalence and incidence given baseline transmission conditions characterised by the annual biting rate and associated mf prevalence, and to understand the impact of ivermectin treatment on the temporal trajectories of OAE prevalence and incidence given programmatic factors such as treatment duration, coverage and frequency. Our simulations suggest that treatment programmes can lead to substantial reductions in both prevalence and incidence of OAE, and in some cases reduce OAE to near-zero cases depending on epidemiological setting, programme duration and whether frequency and coverage can be increased. Our results indicate that in holoendemic areas, characterised by intense transmission and mf prevalence of 80% or greater, 25 years of treatment may be insufficient for elimination of OAE, which can resurge if treatment is stopped prematurely. Therefore, our model could help guide control programmes in terms of treatment duration, coverage and frequency required to reduce or eliminate this important onchocerciasis-associated morbidity.

Next steps: We refer the reviewer to the following (revised) paragraph in the Discussion, where we indicate next steps:

“In holoendemic settings, alternative and/or complementary strategies will likely be required to markedly reduce transmission and protect gains against OAE. Modelling the impact of larviciding (as done for the Sanaga river [29], of which the Mbam is its most important tributary), and/or of other vector control methods such as ‘slash-and-clear’ [30], alone or in combination with a switch to moxidectin MDA would be an important avenue for further work, given the superior efficacy of moxidectin and similar safety profile compared to ivermectin in clinical trials and its potential to accelerate onchocerciasis elimination as shown in modelling studies [31]. Given that our work highlights the contribution to OAE incidence of currently untreated children, moxidectin could help address this gap. Paediatric dose-finding studies seeking to investigate the pharmacokinetics and safety of moxidectin for the treatment of 4–11-year-olds have been completed [32], and modelling studies should be undertaken to investigate the impact of moxidectin MDA on OAE prevalence and incidence compared to ivermectin. Modelling studies should also be undertaken to quantify the impact of supplementing annual CDTI with an extra treatment round targeted at school-age children [33].”

We have also added the following concluding remark to our revised MS by way of providing a link with the original objectives of the study:

“Our study strengthens the argument to include OAE in future iterations of the GBD Study to fully capture the range of onchocerciasis-associated disease sequelae in addition to the currently considered cutaneous and ocular clinical manifestations [3]. To this end, it will be crucial to incorporate OAE-associated premature mortality into EPIONCHO-IBM.”

3.5. Selected specific comments:

Title. The title gives the impression that a general model for OAE was developed which is not true. At least the country needs to be added.

Response by authors to 3.5

Although we have parameterised the model with the dose-response relationship between mf load and risk of developing epilepsy later in life reported by Chesnais et al. 2018 in the Mbam Valley of Cameroon, we have demonstrated that the model can generate outcomes consistent with the relationship between mf prevalence and epilepsy prevalence across several sub-Saharan Africa settings (Figure 4; Response 1.3 and 3.3). Therefore, our model is sufficiently general that our paper does not warrant a change in title or mention of a specific country.

We have added the following text to the Discussion to highlight this:

“In conclusion, integrating OAE into EPIONCHO-IBM, via the dose-response relationship between *O. volvulus* mf load in childhood and risk of developing epilepsy later in life reported in [10], closely reproduces observed prevalence and incidence after 19 years of annual CDTI in the Mbam Valley area of Cameroon [10,18], as well as the relationship between mf prevalence and epilepsy prevalence across several SSA epidemiological settings [7] (Figure 4). This suggests that such dose-relationship captures a fundamental process by which *O. volvulus* infection is associated with epilepsy rather than being an observation specific to the study area, conferring a high degree of generalisability to our modelling framework.”

3.6. Line 16. Donated ivermectin can be used in children 5 years and older, and moxidectin has been evaluated for children 4 years and older. To call for expensive treatment studies in pre-school aged children (aged 3-4) requires a very strong scientific justification. Since these young children are rarely mf positive, significant reduction of the mf reservoir in older subjects may be sufficient to reduce the risk of early development of OAE.

Response by authors to 3.6

Call for expensive treatment studies: We would like to clarify that we do not call for expensive treatment studies in children. Instead, we advocate, given that regulatory approval is being sought for the treatment of 4–11-year-olds with moxidectin (by Medicines Development for Global Health) that modelling studies be conducted to compare the relative impact of ivermectin and moxidectin on both transmission and

associated morbidities such as epilepsy. An additional treatment round in school-aged children has previously been proposed (Colebunders et al. 2022, cited in our MS).

Therefore, we propose that modelling studies be conducted to test the impact of this strategy, precisely one of the 'next steps' with our model. We have clarified this point in the Discussion as already described in Response 3.4.

Young children are rarely mf positive: The studies by Chesnais et al. 2018 and 2020, as well as other epidemiological studies (Basáñez and Boussinesq 1999) show that both mf prevalence and mf loads in children can be substantial in areas with high transmission intensity (e.g. Figure 6 and Figure 7 of Basáñez and Boussinesq 1999), so it is likely that these children are exposed from birth. It is precisely this high exposure at an early age which is considered an important risk factor for the development of epilepsy in later life (Chesnais et al. 2018, 2020). In these areas, treatment of older subjects alone may not be enough to reduce transmission sufficiently to protect these children.

See:

Basáñez MG, Boussinesq M. Population biology of human onchocerciasis. *Philos Trans R Soc Lond B Biol Sci.* **1999**; 354:809–26.

It has been shown that *O. volvulus* infection reduces cellular immune responses to homologous parasite antigens. This "parasite tolerance" can be transmitted from mother to child (Soboslay et al. 1999). In an 18-year follow-up study of approximately 4,000 families in West Africa, children born to *O. volvulus*-infected mothers had four-fold higher odds of becoming *O. volvulus*-infected and also developed high microfilarial loads earlier in life (Kirch et al. 2003). Therefore, children born to *O. volvulus*-infected mothers may be at increased risk of developing a high-level *O. volvulus* infection at a young age, subsequently increasing their risk of developing OAE if not treated with ivermectin.

See:

Soboslay PT, Geiger SM, Drabner B, et al. Prenatal immune priming in onchocerciasis: *Onchocerca volvulus*-specific cellular responsiveness and cytokine production in newborns from infected mothers. *Clin Exp Immunol* **1999**; 117:130–7.

Kirch AK, Duerr HP, Boatin B, et al. Impact of parental onchocerciasis and intensity of transmission on development and persistence of *Onchocerca volvulus* infection in offspring: an 18-year follow-up study. *Parasitology* **2003**; 127:327–35.

3.7. Line 23, 27, 28. Is it necessary to introduce abbreviations like SSA, OCP, or APOC that are only used once or twice in the entire paper?

Response by authors to 3.7

These are not standard abbreviations, but are used in the onchocerciasis literature more generally; therefore, we would like to keep them as readers will likely come across these terms in papers from our group and others.

3.8. Line 28. While previous control programs were mentioned, it would be important to mention the WHO program for the elimination of human onchocerciasis (and potentially OAE). This program is in agreement with the roadmap for neglected tropical diseases of the WHO, is mostly based on mass drug administration and has to ultimate goal of elimination of *O. volvulus* transmission.

Response by authors to 3.8

We thank the reviewer for highlighting this. We have now introduced a reference to the 2021-2030 WHO roadmap on NTDs in the revised Discussion, as follows:

“More generally, our model can help inform onchocerciasis control and elimination programmes aimed to interrupt transmission as proposed by the World Health Organization 2021–2030 Roadmap on NTDs [34] and consequently to eliminate OAE.”

Reference [34] is:

World Health Organization. Ending the neglect to attain the Sustainable Development Goals: a road map for neglected tropical diseases 2021– 2030. Geneva: World Health Organization, 2021. Available at:

<https://www.who.int/publications/i/item/9789240010352>. Accessed 11 May 2024.

3.9. Line 56-64. Although the authors mostly refer to already published data, important background information is missing that should be included in the EPIONCHO model. For example, high rates of epilepsy caused by cysticercosis have been described in parts of Cameroon (Angwafor et al. 2021). How was OAE and cysticercosis associated epilepsy differentiated?

What proportion of epilepsy cases was ascribed to which worm infection in the model? If the authors don't have data on other infections, they should at least mention this additional limitation in their limitation section.

Response by authors to 3.9

Since EPIONCHO-IBM is a transmission model for onchocerciasis, it is not designed to model epilepsy due to taeniasis/cysticercosis. For this purpose, we have developed the EPICYST transmission model for *Taenia solium* epidemiology and control (see Winskill et

al. 2017 PMID: 28183336; Dixon et al. 2019, 2020a, 2020b, 2021, 2022 PMID: 30969966, 32132754, 33077748, 34024358, 35984416, respectively). The papers by Angwafor et al. 2021 refer to the North-West Region of Cameroon, and hence they are not relevant for the Centre Region. Notwithstanding this, we acknowledge that there may be other infectious causes of epilepsy in the Mbam valley, where cysticercosis has been reported (Boussinesq et al. 2002). Particularly, in the village of Bilomo (located in the valley but not in the study area of Chesnais et al. 2018 or 2020), a case-control study was conducted in 1998 (Dongmo et al. 2004) which failed to reveal a statistically significant association between seropositivity for cysticercosis (by ELISA) and epilepsy, with the authors proposing onchocerciasis as a possible explanation for the high epilepsy prevalence found in the village. A subsequent investigation conducted by Siewe-Fodjo et al. 2019, in the Mbam valley, including Nyamongo and Ngongol (in the Chesnais et al. 2018 study) as well as Bilomo, documented that more than 90% of persons with epilepsy fulfilled the criteria for OAE. Nevertheless, we do agree with the reviewer that it is important to better understand the co-distribution of onchocerciasis and cysticercosis in relation to epilepsy in SSA and have made a plea for further studies on this in other publications.

We have revised the text in the section on Limitations as follows:

“Although not all SCE identified by Chesnais et al. [10] would have been OAE cases, a study conducted in the same area and at the same time (in 2017–18) ascertained that 93.2% of person with epilepsy fulfilled the OAE diagnostic criteria [24]. We acknowledge that cysticercosis (another infection associated with epilepsy) has been reported in the Mbam valley [25]. However, a case-control study conducted in the village of Bilomo (located in the valley) failed to reveal a statistically significant association between seropositivity for cysticercosis and epilepsy, with the authors proposing onchocerciasis as an alternative explanation for the high epilepsy prevalence found in the village [26].

More generally, a better understanding and mapping of the co-endemicity of onchocerciasis and taeniasis/cysticercosis in relation to epilepsy in SSA constitutes an important research gap that needs to be urgently addressed [27].”

With new references [25], [26] and [27] added:

25. Boussinesq M, Pion SDS, Demanga-Ngangue, Kamgno J. Relationship between onchocerciasis and epilepsy: a matched case-control study in the Mbam Valley, Republic of Cameroon. *Trans R Soc Trop Med Hyg* **2002**; 96:537–41.
26. Dongmo L, Druet-Cabanac M, Moyou SR, et al. Cysticercose et épilepsie: étude cas-témoins dans la Vallée du Mbam, Cameroun. *Bull Soc Pathol Exot* **2004**; 97:105–8.

27. Otabil KB, Ankrah B, Bart-Plange EJ, et al. Prevalence of epilepsy in the onchocerciasis endemic middle belt of Ghana after 27 years of mass drug administration with ivermectin. *Infect Dis Poverty* **2023**; 12:75.

Furthermore, in certain onchocerciasis-endemic areas in South Sudan, where neurocysticercosis cannot explain the high epilepsy prevalence since no pigs are kept in the area, a similar association between onchocerciasis and epilepsy as in the Mbam valley was observed (Colebunders et al. 2018).

See:

Colebunders R, Carter JY, Olore PC, et al. High prevalence of onchocerciasis-associated epilepsy in villages in Maridi County, Republic of South Sudan: a community-based survey. *Seizure* **2018**; 63:93–101.

3.10. Line 117. How was 'holoendemic' endemicity (80% mf rate) calculated? Children under the age of 3 years are usually mf negative and mf rates in preschool and primary school aged children are usually very low. Migrants from larger cities such as Yaounde are also mostly mf negative. Therefore, it is difficult to achieve a sex/age standardized mf rate of at least 80%.

Response by authors to 3.10

Holoendemicity was modelled by increasing the vector biting rate according to the (strongly non-linear) relationship between ABR and mf prevalence that characterises EPIONCHO-IBM (Hamley et al. 2019) and that allows us to model onchocerciasis transmission for any epidemiological setting (see also Response 1.3). In holoendemic foci, transmission is very intense, and children can be exposed from birth, developing sizeable mf prevalence and mf loads by the time they reach pre-school and school age (see also Response 3.6, where we give examples of this from other epidemiological studies). Furthermore, we have recently conducted a systematic review and meta-analysis of published studies reporting on the status of onchocerciasis elimination of transmission, and have found that baseline holoendemicity is statistically significantly associated with increased risk of ongoing transmission under long-term ivermectin MDA. This underscores the importance of understanding the impact of interventions in initially holoendemic settings on OAE prevalence and incidence.

See:

Mutono N, Basáñez MG, James A, et al. Elimination of transmission of onchocerciasis (river blindness) with long-term ivermectin mass drug administration with or without

vector control in sub-Saharan Africa: a systematic review and meta-analysis. *Lancet Glob Health*. **2024**; 12:e771-e782.

3.11. Line 139. The table should not only include the publication year, but the year when data were collected and the exact villages where data were collected.

Response by authors to 3.11

We have now added this information as a footnote to Table 1, as follows:

“The data presented in Pion et al. [15] were collected in 1991–1993 in the areas of Bitang, Yambassa and Yébékolo (see Supplementary Information Figure S1, which reproduces the study area of [10]; the data presented in Barbazan et al. [16] were collected in 1993–1994 in Ngoro and Bokito (also in Figure S1).”

3.12. Line 292 The model predicts that 3-4 year old children contribute 10-12% of the total OAE. What was the mf rate in this age group? Or at least what was the exposure to *Simulium* bites? Children in this age group are usually not working or playing close to the rivers with high annual biting rates.

Response by authors to 3.12

Since in our model it is the mf load (rather than the prevalence) which drives the probability of developing epilepsy, we provide the referee with results for mf load. According to the simulations presented in Figure 5, children aged 3–4 years in the hyperendemic and holoendemic modelled populations would have a mean mf load of, respectively, 3–4 and 6–10 microfilariae/skin snip. These fall in the second and third binned categories of Figure 1, which correspond to probabilities of developing epilepsy later in life of 4% to 7% (see also Table S2, reproduced from Chesnais et al. 2018), ultimately leading to our estimation that these children contribute to 10-12% of the overall OAE incidence. Regarding exposure, Supplementary Information Figure S2 depicts the age- and sex-specific exposure functions of EPIONCHO-IBM. These exposure functions were derived by fitting an age- and sex-structured precursor of the model to age and sex profiles of mf infection data from Cameroon (Filipe et al. 2005) as described in the Supplementary Information (see also Response 2.6), indicating that exposure would not be negligible at birth.

Although children in this age group may not be working or playing close to rivers with high annual biting rates, it is not uncommon for toddlers to be carried on their mothers' backs when they themselves go to locations near rivers or engage in farming activities. Furthermore, children from *O. volvulus*-infected mothers have a substantially higher risk

to become infected and also acquire infection earlier in life, developing higher mf loads (Kirch et al. 2003; see Response 3.6). As indicated in Response to 3.6, young children with high mf loads are key in the dose-response relationship. In areas of high transmission, children aged 3–9 years may have mf prevalence in excess of 30% in hyperendemic areas and greater than 60% in holoendemic areas (Figure 6 of Basáñez & Boussinesq 1999), lending support to the notion that children become infected early in life in areas of high transmission intensity.

Reviewer #3 (Remarks on code availability):

Although I worked together with modellers before, I am not a computer modeler and I did not try to install and run the code.

We thank the reviewer for the in-depth review despite not being a modeller.

REVIEWERS' COMMENTS

Reviewer #1 (Remarks to the Author):

I am Happy with the responses provided by the authors.

Reviewer #3 (Remarks to the Author):

The authors prepared an extensive rebuttal letter and replied in detail to the reviewers comments. While they made some important changes, the reviewer still believes that they overstate the relevance of this model that uses incomplete, published data (no epilepsy in 1991-93, no parasitology in 2017) from a single area in Cameroon. With regard to the comment 3.3 the reviewer does not agree that the "results suggest that the dose-response relation ship reported by Chesnais et al. 2018 is fundamental to the process by which onchocerciasis is associated with epilepsy more general". For example the epidemiology of onchocerciasis in Cameroon is widely known to be very different from the epidemiology of *O. volvulus* transmitted by *Simulium neavei* in Uganda where historically epilepsy associated with oncho was first described. The authors failed to provide any evidence that their model is applicable for all OAE in Africa.

Responses to Reviewers' Comments (NCOMMS-24-16977A)

Reviewer #1 (Remarks to the Author):

I am Happy with the responses provided by the authors.

Response to Reviewer #1

We are pleased the reviewer has been satisfied by our revisions and would like to again thank them for their insightful comments which considerably strengthened our manuscript.

Reviewer #3 (Remarks to the Author):

3.1. The authors prepared an extensive rebuttal letter and replied in detail to the reviewers comments. While they made some important changes, the reviewer still believes that they overstate the relevance of this model that uses incomplete, published data (no epilepsy in 1991-93, no parasitology in 2017) from a single area in Cameroon.

Response to 3.1:

We are pleased to learn that the reviewer found our response letter to be comprehensive. The parasitological and epilepsy data from Chesnais et al. 2018 pertain to a retrospective cohort of children (aged 5-10 years) identified 25 years later to investigate the extent to which the magnitude of microfilarial load harboured during childhood related to the probability of developing epilepsy later in life. As such, only parasitological data were collected in 1991-1993 and epilepsy data in 2017, which helped establish a dose-response and temporal relationship between *O. volvulus* infection intensity in the past and subsequent development of epilepsy. As the authors reconstructed a cohort of 729 individuals, each individual had a record of parasitological and epilepsy data and, therefore, the data were not incomplete. We agree with the reviewer that the separate collection of parasitological and epilepsy data requires clarification and have modified the text in the Abstract as follows:

“Retrospective cohort studies in Cameroon found an association between *Onchocerca volvulus* microfilarial load in childhood (measured in 1991–1993) and risk of developing epilepsy later in life (measured in 2017).”

3.2. With regard to the comment 3.3 the reviewer does not agree that the "results suggest that the dose-response relationship reported by Chesnais et al. 2018 is fundamental to the process by which onchocerciasis is associated with epilepsy more generally". For example, the epidemiology of onchocerciasis in Cameroon is widely known to be very different from the epidemiology of *O. volvulus* transmitted by *Simulium neavei* in Uganda where historically epilepsy associated with oncho was first described. The authors failed to provide any evidence that their model is applicable for all OAE in Africa.

Response to 3.2:

We thank the reviewer for raising this important point. Our model captures well the dose-response relationship between *O. volvulus* microfilarial load acquired during childhood and the probability of developing epilepsy later in life, and it is this we believe is a fundamental process by which onchocerciasis is associated with epilepsy. As is also seen in Figure 4B, the relationship between microfilarial prevalence and epilepsy prevalence from two studies in western Uganda (Kipp et al., 1994 – black circles – and Kaiser et al., 1996 – white triangles –) is well captured by our model predictions for three levels of exposure heterogeneity. We acknowledge, however, that due to entomological differences between the vector species in the Mbam Valley (*S. damnosum s.l.*) and that found in western Uganda (*S. neavei*), we cannot rule out that differing transmission characteristics may influence OAE incidence and prevalence patterns.

We have added text to the Discussion to address this point:

“Besides, our model was parameterised with *Simulium damnosum* sensu lato as the vector, but in Uganda, where OAE was first reported in Africa [22], transmission was due to *S. neavei* [23]. Therefore, it cannot be ruled out that the transmission characteristics of different vector groups may influence OAE patterns [...].”

And we have modified the abstract to clarify that our model was parameterised with data concerning *S. damnosum s.l.*:

“We parameterised and integrated this relationship (across children aged 3–15 years) into the previously published, stochastic transmission model, EPIONCHO-IBM, for *Simulium damnosum* sensu lato-transmitted onchocerciasis.”